

# Optimizing Methane Emission Source Localization in Oil and Gas Facilities Using Lagrangian Stochastic Models and Gradient-Based Detection Tools

Afshan Khaleghi[1,2], Mathias Göckede[3], Nicholas Nickerson[4], David Risk[1]

5   [1] Department of Earth and Environmental Sciences St. Francis Xavier University Antigonish, Nova Scotia, Canada
[2] Department of Process Engineering, Memorial University of Newfoundland, Newfoundland and Labrador, Canada
[3] Max Planck Institute for Biogeochemistry, Jena, Germany
10  [4] Eosense Inc, Nova Scotia, Canada

*Correspondence to*: Afshan Khaleghi (akhaleghi@stfx.ca)

**Abstract.** Oil and gas facilities are responsible for approximately 35% of global emissions of methane (CH₄), a potent greenhouse gas, posing significant environmental and regulatory challenges. While Continuous Emission Monitoring Systems (CEMS) are widely implemented to track real-time emissions, 15   their effectiveness in localizing specific CH₄ emission sources remains limited, particularly under complex environmental conditions. This study integrates CEMS technologies with a Lagrangian Stochastic Back-Trajectory Model, along with an automated Gradient Indicator (GI) tool to improve methane source localization accuracy in oil and gas settings. The model was then applied to a real-world gas distribution site to validate its performance in accurately localizing methane emissions under 20   operational conditions. Using synthetic data simulations, we evaluated the performance of this integrated system under various atmospheric stability conditions, sensor-source height differences, and source proximity. Our results indicate that this combined approach significantly enhances localization performance, achieving a 90% probability of detection (POD) within a 25–75 meter source-sensor distance under optimal conditions. However, detection performance varied across configurations, with 25   false positive rates (FPF) ranging from 22% to 86%, and localization accuracy (LA) ranging from 14% to 78%, depending on atmospheric stability, source-sensor geometry, and height differences. The Localization Accuracy (LA) improves when sensor placements are exactly downwind of the emission sources (alignment). The system meets Canadian regulatory requirements for CEMS applications by maintaining localization accuracy above 90% for unstable and slightly neutral atmospheric conditions, 30   ensuring that emissions are correctly attributed. However, neutral atmospheric conditions and large height differentials between sensors and sources reduce localization accuracy, making optimized sensor configurations important. Findings of the research can be useful for upgrading CEMS systems and help them to overcome some difficulties from regulations associated with methane emission reporting and mitigation efforts.

**Keywords:** Methane emissions, source localization, oil and gas, Gradient Indicator tool



## 1 Introduction

Methane (CH₄) is much more powerful in trapping heat in the atmosphere than carbon dioxide (CO₂). These emissions, as a result from anthropogenic sources and natural, contributed to this, cause the rise of global temperature by about 0.3° to 0.8°C within 2010-2019 compared to that during 1850-1900, states
the IPCC 2023. Canada has been forward-thinking on this topic and has moved at a provincial and federal level, such as by aligning with the international Global Methane Pledge, which was set to reduce methane (CH₄) emissions by 30% before 2030. The oil and gas sector contributes 13% of total methane emissions in Canada and therefore provides significant potential for targeted cuts. Canada has set a national target of cutting these emissions by 75% from 2012 levels by 2030 as part of its commitment to the Global
Methane Pledge (Government of Canada, 2024). This requires identifying the source of CH₄ emissions and mitigating it in order to achieve these global and national targets. Emission estimates from the oil and gas industry have shown underreporting by up to 1.5 times (Chan et al., 2020; Chen et al., 2022; Conrad et al., 2023; MacKay et al., 2021; MacKay et al., 2024; Seymour et al., 2023; Vogt et al., 2022). Understanding the contribution of different sources is essential for both detecting and managing leaks and
improving future inventory estimates.
While combining multiple techniques can enhance source localization, stationary sensors still struggle to detect and differentiate emissions from multiple sources. Continuous Emission Monitoring Systems (CEMS), as noted by Jia et al. (2023), primarily measure ambient methane concentrations rather than direct emission rates, making it difficult to pinpoint sources when multiple emitters are present. Wang et
al. (2022) highlighted that CEMS's 1-minute averaging frequency improves the chances of capturing transient emissions lasting only 24 hours. However, additional operator input is often necessary to refine accuracy.
Chen et al. (2022) found that intermittent large emissions can go undetected by CEMS networks, creating critical data gaps that hinder precise source localization. As pointed out by Daniels et al. (2022), even
wind speed and direction are problematic for emission source estimation. Bell et al. (2022) also observed that variability in sensor performance and environmental conditions may further constrain methane quantification accuracy. Methods in which dense networks of stationary sensors have been used have, to date, made precise identification of emission sources quite difficult; therefore, substantial improvement in methodology is needed.
Latest studies have demonstrated the efficacy of CEMS under both controlled and practical circumstances. Chen et al. (2023a) assessed the detection efficiency of the CEMS under diverse site-specific conditions and recorded considerable fluctuation in accuracy, hence heightening the likelihood of false negatives in source detection. Daniels et al. 2024 estimated single-source methane emission through spatial monitoring and found that the placement of sensors and atmospheric conditions are very
critical to detector reliability. Jia et al. 2023 evaluated Gaussian plume and puff models for methane dispersion and determined that the rudimentary back-trajectory methods used in most commercial CEMS are likely to be inadequate for complex emission scenarios.
Experimental CEMS solution research still points to performance limitations and areas for improvement. Bell et al. (2023) and Ilonze et al. (2024), in a single-blind controlled test, confirm that while some
continuous monitoring solutions perform well under ideal conditions, field applications still suffer from false positives and high uncertainty. Daniels et al. (2023) explored how continuous monitoring data can



be reconciled with regional methane inventories, noting that integration of both bottom-up and top-down approaches has the potential to decrease overall uncertainties. Cardoso-Saldaña (2023) proved that a tiered leak detection program significantly increases the effectiveness of emission reduction when combined with continuous monitoring.

Most of these works emphasize the optimization of sensor placement to avoid false detection and ensure coverage. However, all CEMS algorithms remain proprietary commercially, with the probable use of some form of back-trajectory mapping for the most part, further limiting true attribution. This integration with CEMS offers exciting opportunities to take AI-driven atmospheric models and multi-sensor fusion methods toward the detection and mitigation of methane emissions in real time.

Lagrangian Particle Dispersion Models (LPDMs) have been shown to be useful in estimating $CH_4$ emissions from possible sources without the need to pinpoint source locations. Originally designed for natural sources, LPDMs such as the Stochastic Time-Inverted Lagrangian Transport model (STILT) (Lin et al., 2003), $CO_2$ Methane Transport COMET (Vermeulen et al., 1999), and Global Eulerian-Lagrangian Coupled Atmospheric (GELCA) (Ganshin et al., 2012) have proven effective in quantifying emissions from oil and gas facilities at regional scales (Lin et al., 2021). While both flux and concentration footprints identify source regions influencing a measurement, flux footprint approaches only trace back trajectories to the first flux source (emission rate), and consider also the vertical particle movement when passing the sensor locations. Concentration footprints, on the other hand, allow to trace contributions across multiple sources upwind of the sensor, and consider only the number of particles passing the sensor location, not their vertical movement. As a consequence, flux footprints tend to be smaller than concentration footprints, and dimensions of concentration footprints can cover very large areas, depending on how far back in time trajectories will be followed.

Recent advances in the field have demonstrated the effectiveness of the concentration footprint technique in a wide variety of applications. Sintermann et al. (2011) illustrated the use of backward Lagrangian stochastic modelling in conjunction with optical Fourier Transform Infrared (FTIR) systems in attributing ammonia emissions from agricultural practices, showing the versatility of this approach. Similarly, *Yu et al.* (2021) presented important seasonal changes in concentration footprints from regional model STILT for surface $CO_2$ exchanges. These studies underscore the potential of this method in terms of the accuracy of data about atmospheric processes and emissions. *Levin et al.* (2021) discussed the lack of traditional methods for capturing point sources in emissions and emphasize the requirement for robust concentration footprint analyses in developing improved inventories of greenhouse gases. That makes the Lagrangian concentration footprint method relevant for enhancing emissions assessments, especially within oil and gas industry.

In this study, we applied Tool for Emission Response and Retrieval for Flux Exploration in R (TERRAFEX), a concentration Lagrangian back-trajectory method at site level , developed by *Göckede et al.* (2004, 2006), to create source contribution maps upwind of the sensor location. A Gradient Indicator (GI) tool was designed to identify the location of point source emissions (localization) using TERRAFEX maps. Together, TERRAFEX and the GI represent a model system to automatically establish emission source locations. The study aims to explore the sensitivities of the model system to determine the optimal sensor configuration, including its relative height compared to oil and gas infrastructure. By emphasizing emissions at the facility level, the approach enables increased localization of sources and enhances the methods for mitigative practices. Synthetic data have been used within the study to understand the impacts





of parameters related to source-sensor height, atmospheric stability class, and relative distance between
the source and sensor on the sensitivity of the source localization. The methodology has been applied
to a real gas distribution site for verification of the capability of the methodology in operational
conditions       and demonstration at       a       real       site       for source location       with       high
accuracy. This paper further enhances the       accuracy       and       reliability       of       the       CEMS       systems
by improving source localization at a facility level and gives insight into refinement of other dispersion
methods.

## 2 Method development

### 2.1 Lagrangian footprint model

The framework to couple Lagrangian backward source weight calculations with land cover characteristics
for data quality assessment (TERRAFEX) was developed by *Göckede et al.* (2006). The footprint model
used therein was primarily applied for flux footprint calculations for Eddy Covariance (EC) tower studies,
but with minor modifications can also be used to calculate concentration footprints and adopt the model
for application with concentration sensors. The main difference between flux and concentration footprint
model versions is that for fluxes, the vertical wind speed of particles crossing the sensor location is
considered, while for concentrations, only the total number of particles matters. In other words, in a flux
footprint, the opposing movements require deducting the net flow of downward particles from those
moving upward, whereas, for a concentration footprint, all passing particles are summed aside from their
direction (Aubinet et al. 2012). A flux footprint therefore separates and balances upward and downward
directed flux contributions, while in concentration footprint, all emissions have positive contributions and
contain only sources.
The Lagrangian footprint method integrated into TERRAFEX simulates particle trajectories forward in
time, but based on the inverted plume assumption, the same transport and dispersion patterns can be
assumed away from the sensor point (Aubinet et al. 2012), upwind towards the source(s). Many footprint
methods are computationally intensive, especially for long-duration high-frequency simulations.
TERRAFEX offers some simplifications that significantly reduce computational intensity for large
datasets, mainly by using pre-calculated source weight function tables for user-defined classes of
measurement height, Obukhov length, and roughness length. The basic principle is that first, the source
weight function is selected from the pre-calculated footprint tables, then this function is projected onto
the discrete grid covering the study area, and then the weights per pixel are multiplied with the target
parameter, in this case, the $CH_4$ concentration. For each timestep of measurement, TERRAFEX generates
a concentration source contribution map upwind of the sensor's location. The output of the simulation
over time is the cumulative sum of the maps produced at the individual timesteps. Using a stationary
sensor, it is important that this accumulation covers a range of wind conditions (wind direction, stability).
Since the wind direction deviation from average will be limited for short term measurements, short
campaigns can bring uncertainty to the resulting commutative map.
This study used different sets of synthetic data with TERRAFEX to understand our ability to derive source
locations from its concentration footprint map outputs. For our study, the TERRAFEX model and all



other scripts were written and executed in R (R Core Team, 2024) operating on local UNIX or OSX machines.

## 2.2 Gradient indicator tool

In the concentration source contribution maps, we would expect an increase from background to higher values in case the footprint overlaps with regions where $CH_4$ sources exist. To automatically identify these locations, we developed a gradient indicator (GI) tool. Finding gradients relies on $CH_4$ sources being represented on a concentration source contribution map as a cone-shaped feature, decreasing in concentration with distance along the wind path, from a concentration maxima near the primary position
of the source.
The main loop of the GI tool consists of a ranking method to flag the highest concentration points using different gradient lengths (cell numbers) starting from one. This is an assumption in lack of information from the source location (meaning that each concentration point is being flagged as high compared to itself). As shown in Fig. 1, the gradient length will increase incrementally with each loop and will proceed
in both x and y directions. The loop continues until at least one of the remaining points on the matrix surpasses the top 5% $CH_4$ concentration cells on the original matrix. The reoccurring gradient loop counts the persistence of points with different lengths and will only keep the high persistence points from the TERRAFEX contribution source maps.
These points can then be filtered further to only keep those that are within the limits of known features
that could be emitting, for example, oil and gas well-pads. We filter data for wind directions associated with the top 90% of the highest wind speeds, considering air stability class for enhancing the localization. Using these filtered parameters, it generates a map that displays the filtered points representing the result of the localization process.





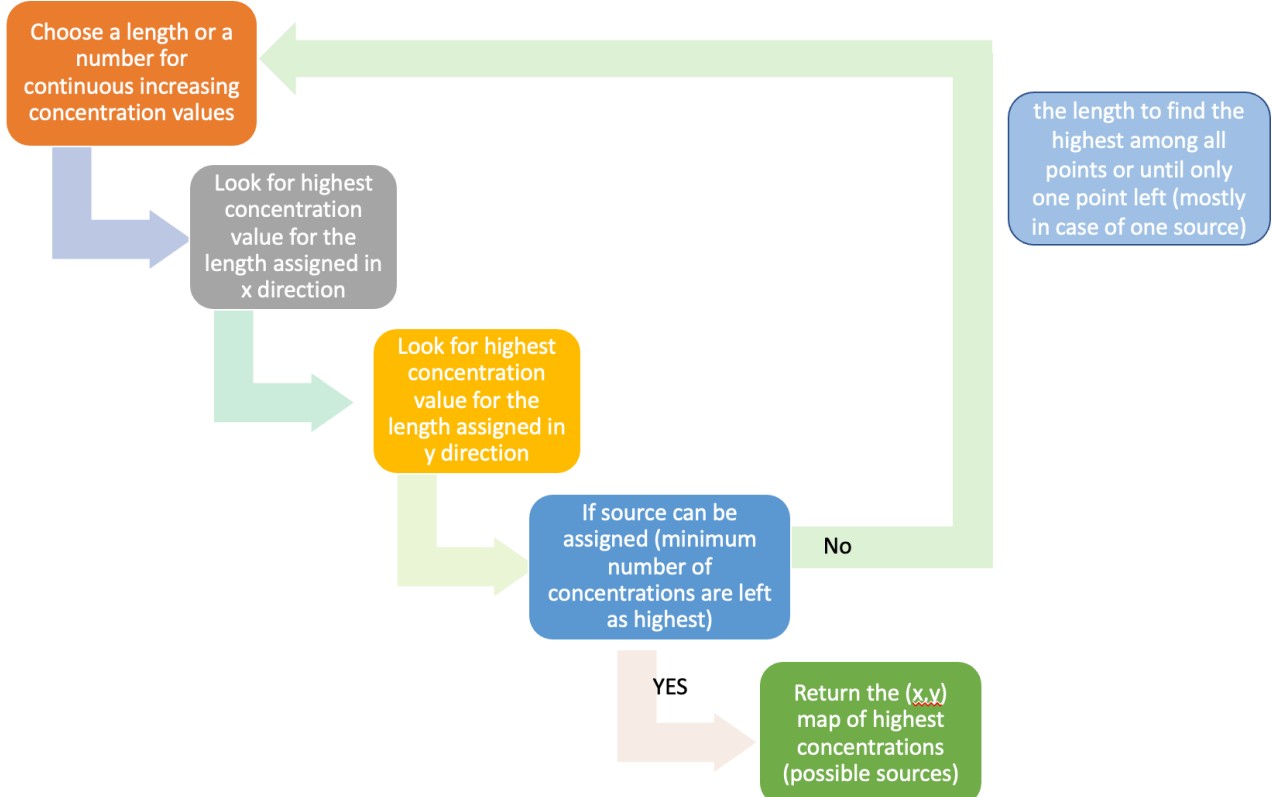

**Figure 1: The workflow of the GI tool begins by selecting a starting value of 1 for the gradient length, or a higher value if information about the source strength or approximate location is available. Since the exact locations are unknown, a gradient length of 1 is typically the default. The next step involves assigning the source (represented by the dark blue box) as long as there are still cells remaining with the highest 5% concentration (within the 95% confidence interval).**

In reality, finding the exact location of source(s) upwind of the sensor requires information about the stability class of air, wind data, number of sources, and number of the sensor(s). There is a high probability of artifacts in the final maps. Artifacts may, at times, arise due to poor variation in the wind direction inside the time series or when the length of the dataset is not long. One thing to bear in mind while using a gradient length indicator is that source location approximation would, at best, be able to be done by it. Also, strong hotspots can easily make the algorithm mark the same persistent points many times with iteration. It must be kept in mind that the gradient length indicator can only provide an approximation of source locations. Due to the presence of strong hotspots, the algorithm might identify the same persistent points more than once during iterations. Considering the presence of significant hotspots, the algorithm may repeatedly identify the same persistent points during the iteration process. To minimize processing time and enhance efficiency, the loop is designed to terminate once the same hotspots are consistently detected. For this purpose, the process will stop when at least one of the remaining matrix points surpasses the top 5% of CH4 concentration levels in the original matrix.



## 2.3 Sensitivity analysis

The main goal of this paper is to understand the positional accuracy, precision, and uncertainty that would
200   come from the bundled use of TERRAFEX, with the addition of a GI tool to interpret its concentration
contribution maps. Here, we built a large matrix sensitivity test using synthetic data where emission
locations are known from forward simulations driven by a Gaussian plume model. The use of synthetic
data allows us to evaluate the effects of sensor placement, source location, and atmospheric conditions in
a controlled manner across a parameter space much broader than we could find in real-world testing.
205   We created our own time-series and $CH_4$ concentration simulation maps using forward GDM modelling,
with inputs including atmospheric stability class, emission source location(s), rate of $CH_4$ emissions, and
background levels. We used real wind data from a campaign in Weyburn, Alberta during December of
2015 in an almost flat terrain with different stability classes of A (highly unstable) to D (neutral). The
data was measured with a Gill Ultrasonic Anemometer for measuring wind speed and direction data with
a precision of 3% route mean square error (RMSE) for wind speed and ±3° for wind direction at a
stationary sensor so that meteorological conditions reflected reality. The data was measured at various
times of the day, but a 20-minute time series was selected for each stability class to represent real
measurement scenarios in an oil and gas facility. The Gaussian model calculates emissions concentration
that can be captured while measuring at any point downwind using the following approach:

$$C(x, y, z) = \frac{Q}{2\pi\sigma_y\sigma_z u} exp\left(-\frac{y^2}{2\sigma_y^2}\right)\left[exp\left(-\frac{(z-h)^2}{2\sigma_z^2}\right) exp\left(-\frac{(z+h)^2}{2\sigma_z^2}\right)\right] \qquad\qquad \text{Eq. 1}$$

Where C is the excess concentration of the chosen substance at point (x, y, z) downwind of the source in
$g/m^3$ (measured minus background), u is the wind speed in m/s, Q is the emission rate from the source in
g/s at $(x_0, y_0, h)$ , considering that $z + h$ in m is the plume rise from the ground (source height plus plume
rise) and $\sigma_y$ and $\sigma_z$ in m are horizontal and vertical dispersion coefficients. Given data limitations,
$\sigma_y$ and $\sigma_z$ were estimated using predefined values for Pasquill-Gifford stability classes A–D, following
Turner (1970) and consistent and aligned with TERRAFEX model requirements and limitations. In all
cases, we used a 41 $m^3$/day source release rate which represent an average from all possible sources of
emission in a well pad including compressors and tanks based on values form Duren et al. (2019).

The wind measurement and simulation time-series length were ~20 minutes for each set. Three
configurations for the source(s) and sensor locations on a well-pad were designed to account for possible
field complications. Figure 2 illustrates the different source(s) and sensor placement scenarios in a well
pad with a 100 m diameter. For involving multiple sources, a combination of two source locations was
chosen from the aligned source with the centreline, a source with a 45° angle towards the north, and/or
the other at a 45° angle towards the south. The three main configurations are: 1) both sources aligned with
the main wind flow toward the sensor, 2) both sources angled with respect to the wind direction, and 3)
one source aligned while the other is angled. Each configuration includes ten different relative source
placements. These three major configurations were selected to represent a wide variety of realistic and
challenging field conditions that sensors may face during emission detection tasks. Each configuration
introduces different complexities in how emission plumes overlap and interact with sensors, testing the
robustness of the Lagrangian model and sensor placement strategies.



The first configuration, in which both sources are aligned with the wind direction, creates scenarios where one plume may obscure the other. This setup mimics conditions on well pads or industrial sites, where
aligned sources can lead to plume shadowing—posing challenges to source differentiation. The second configuration, with both sources angled away from the centerline, tests the model's ability to resolve spatially separated plumes, a common field scenario when sources are scattered. The third setup combines these challenges by placing one source aligned and one angled, leading to partial plume overlap and elongation—introducing further complexity in localization. In all scenarios, the relative height between
source(s) and sensor(s) was initially held constant to isolate spatial factors. These configurations were chosen to reflect conditions where sensors are not optimally placed yet are still required to distinguish and localize multiple sources.

To further evaluate model performance under diverse spatial constraints, the three dual-source scenarios were simulated across four atmospheric stability classes, yielding 120 baseline cases. This was expanded
by varying sensor positions across ten locations ranging from 0 to 100 meters downwind of the well pad, producing an additional 120 scenarios to test the sensitivity of localization to horizontal sensor placement. Finally, 216 additional simulations were generated by fixing the sensor at one location and varying both the source heights (2, 4, and 6 m) and sensor heights (1 to 6 m), again across all source combinations and stability classes. In total, 456 synthetic dispersion cases were generated. This structured dataset allowed
for a systematic evaluation of detection performance under conditions that vary both laterally and vertically, and across different atmospheric transport regimes.

Although other geometries could be considered, these configurations represent a range of source-sensor relationships in a configuration of increasing difficulty. Previous literature highlights plume over shadowing discussed by Pangs et al. (2023) and Griffin et al. (2020), angular detection challenges
discussed by Chen et al. (2023b), and sensor blind spots discussed by Andrews et al. (2023); these test cases are not only realistic but unavoidable to test the model's performance in various environments of detection.

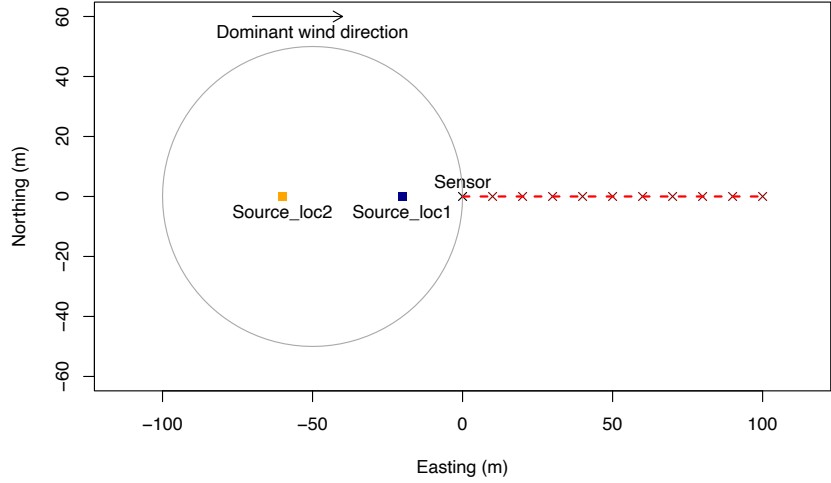




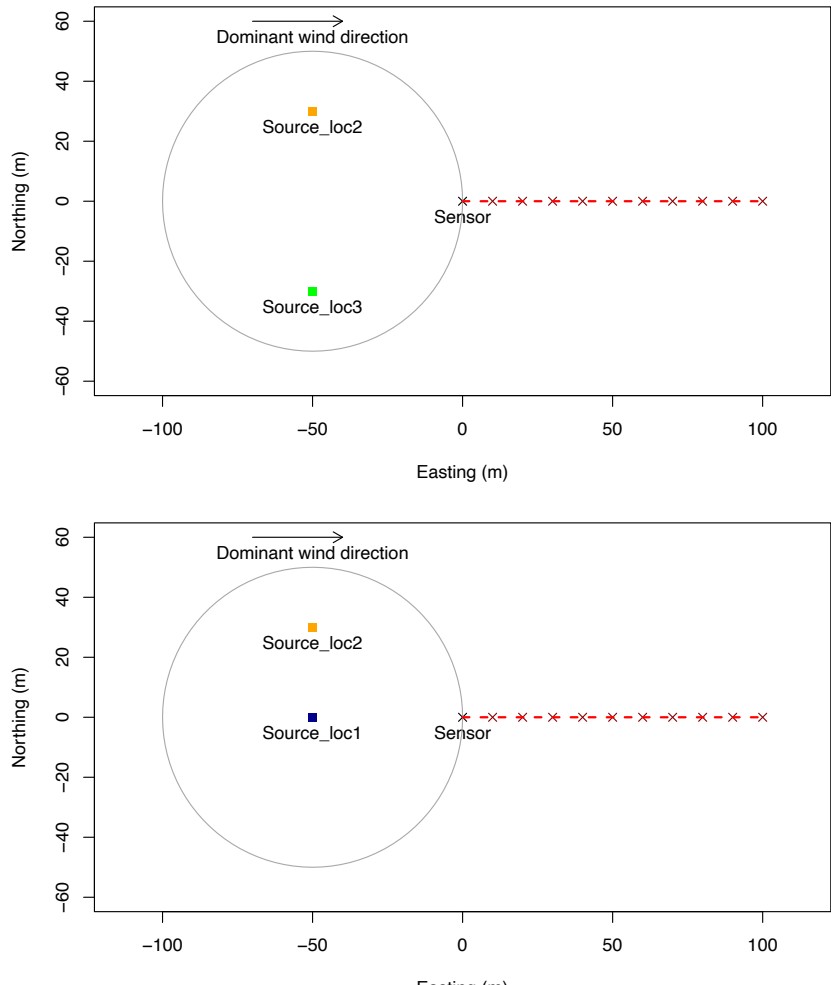

**Figure 2: Location of source(s) and sensor(s) in a well-pad with a 100 m diameter. A combination of two sources with one sensor was used for each scenario. The dashed red line represents different the alignment of sensor while getting away from the edge of well-pad.**

As illustrated in Figure 2, three different sets of analyses were performed to find the best source configuration, sensor relative distance, and the number of sensors needed to find the source(s) location based on the synthetic data from forward Gaussian dispersion model. Overall, 200 data sets were made based on changing the location of sources and sensor for different scenarios. Figure 3 shows a synthetic $CH_4$ concentration time series at one sensor downwind of one-point source in the well-pad within four stability classes.



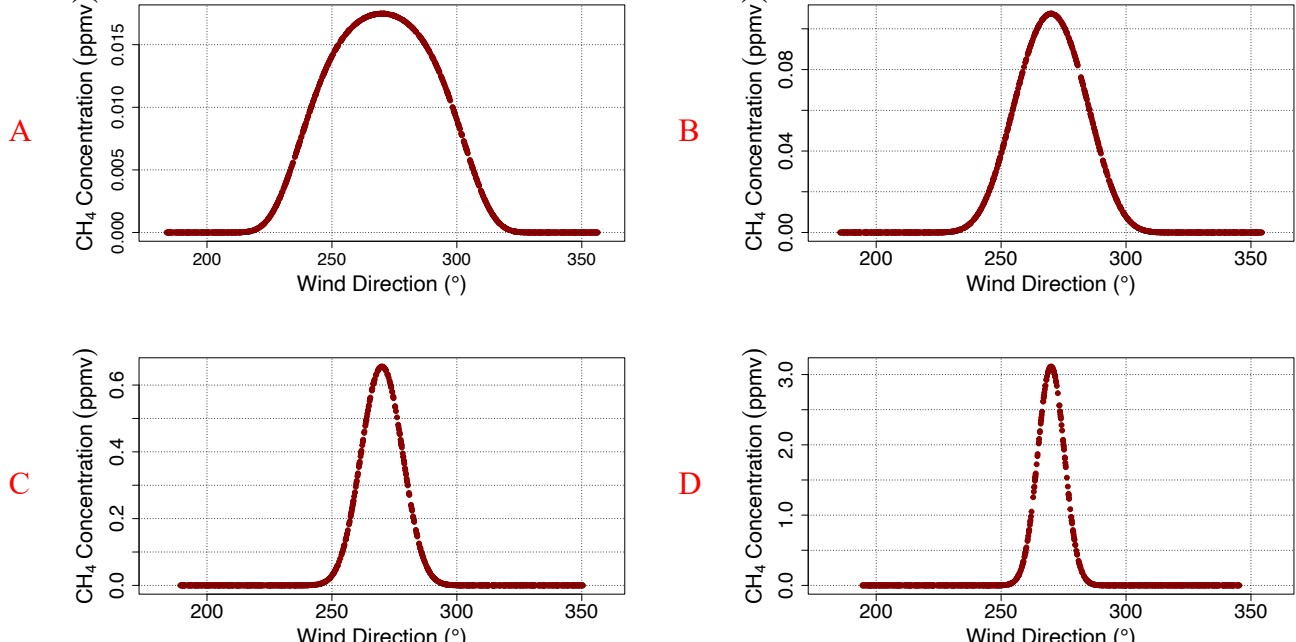

**Figure 3: Simulated CH₄ concentrations (ppmv) as a function of wind direction at a fixed downwind receptor, based on a point source emission under varying wind conditions. The plume is shaped by atmospheric dispersion governed by a single Pasquill-Gifford stability class. Realistic wind variability was used to model transport and dilution effects.**

The method's sensitivity was examined by varying parameters based on the limitations of TERRAFEX. Different scenarios for source locations were created based on eight relative distances between the source(s) for each of the three source positioning scenarios in Figure 2, resulting in 24 cases for which we could characterize the uncertainty of locating known sources with the GI tool. These scenarios were then combined with ten possible sensor locations to derive an estimate of sensitivity to the sensor location. The sensor location was moved away from the edge of the well pad in 10 m increments up to 100 m, resulting in 30 cases. The same three source combinations were used for relative height, selecting three different source heights (2 m, 4 m, 6 m) and six sensor heights, totalling 18 cases. Each case was modelled for four different stability classes of atmosphere (A, B, C, D). The Monin-Obukhov length was used instead of stability class in the application of TERRAFEX and for class A, B, C, and D, it was assumed to be -2, -4, -12 and -1000 respectively based on a table presented by Leelőssy et al. 2014. To account for the real-world scenarios, data was examined with and without adding noise, creating 456 sets of synthetic data. The outline of this paper is shown in Figure 4.





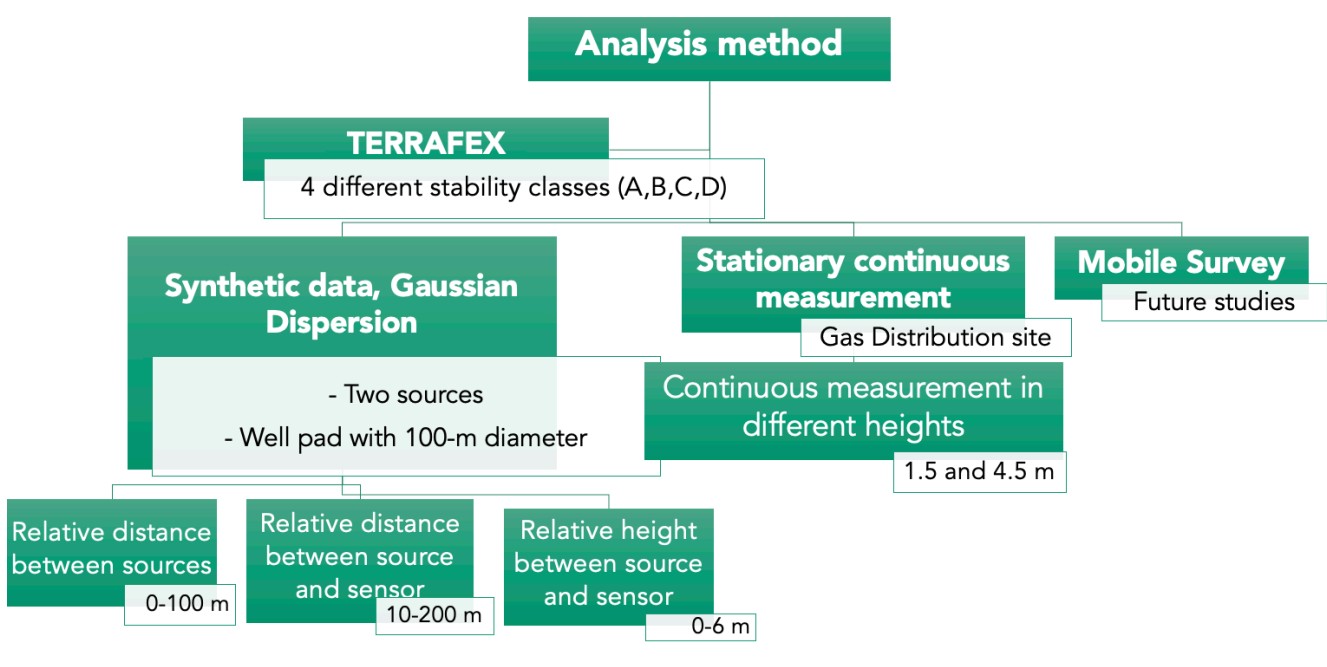


**Figure 4: A flow chart showing steps for stationary source/sensor configuration and validation based on combining TERRAFEX maps and gradient indicator tool to localize the emission sources on a well pad or any oil and gas facility.**

## 2.4 Detection performance assessment

We followed a locational evaluation method developed by the University of Colorado Methane Emissions
Technology Evaluation Center (METEC). The method was used to design controlled release studies for evaluating the performance of different detection and quantification techniques for oil and gas LDAR programs. The most important term used here is Probability of Detection (POD), which is a measure of the effectiveness of detection. The most significant term used herein is Probability of Detection, or POD, which is a measure of effectiveness of detection. Though there are several levels, Bell et al. 2023, relied
on a 90% POD for pointing out the locations for sources using the CEMS network for the minimum threshold, which indeed has been referred to in regulations (ECCC, 2023; EPA, 2024). Here, we applied the same approach, but we defined a criterion for localization limit distance. This study assumed that sources detected within 10 m of their true location were considered True Positive (TP). Conversely, if an actual source is present but not detected within the domain, it is considered a False Negative (FN). Areas
correctly identified as containing no sources are classified as True Negatives (TN) and a False Positive (FP) occurs when the detection system identifies a source as emitting methane when in fact no emission is present at that location. Since the measurement domain is about 200 m, the 10 m buffer represents approximately 5% uncertainty in spatial location. For detected sources where the predicted location is outside the 10 m buffer, we classify these as True Negatives (TN). Knowing this, we can define the
method's effectiveness considering each scenario's criteria and sensitivity. For each case, the POD would be compared within different stability classes, including the average relative distance and the relative height between sources and sensors. The POD can be calculated using:



$$POD = \frac{n_{TP}}{n_{TP}+n_{FN}} \qquad\qquad \text{Eq. 2}$$

where $n_{TP}$ is the number of TP representing cases where an emitting source is correctly detected and $n_{FN}$
is the number of False Negative (FN) representing cases where an actual emitting source is missed by the detection system. For each scenario, False Positive Fraction (FPF), False Negative Fraction (FNF), and Localization Accuracy (LA), which is more related to the accuracy of detection, will be calculated as follows:

$$FPF = \frac{n_{FP}}{n_{FP}+n_{TN}} \qquad\qquad \text{Eq. 3}$$

$$FNF = \frac{n_{FN}}{n_{C}} \qquad\qquad \text{Eq. 4}$$

$$LA = \frac{n_{TP}}{n_{TP}+n_{FP}} \qquad\qquad \text{Eq. 5}$$

where $n_{TP}$ is the number of TP, $n_{FN}$ is the number of FN. $n_{C}$ is the total number of cases in each scenario and $n_{FP}$ is the number of False Positives (FP).

**2.5 Case study**

This method was applied to a CEMS station inside a gas distribution site in Canada as shown in Figure 5. There were four potential fugitive emission sources, and $CH_4$ was measured using an Axetris LGD Compact-A $CH_4$ with 0.01 ppm precision at 2 Hz frequency. The wind was measured using a Decagon DS-2 Sonic Anemometer with 1° and 0.01 m s$^{-1}$ precision for Wind direction and speed, respectively. Measurement height from the stationary sensor was changed from 2 to 4.5 m after one week of
measurement. During this campaign, data was recorded from June 24 to July 8, 2020. This data was used to test the results of sensor configuration analysis.



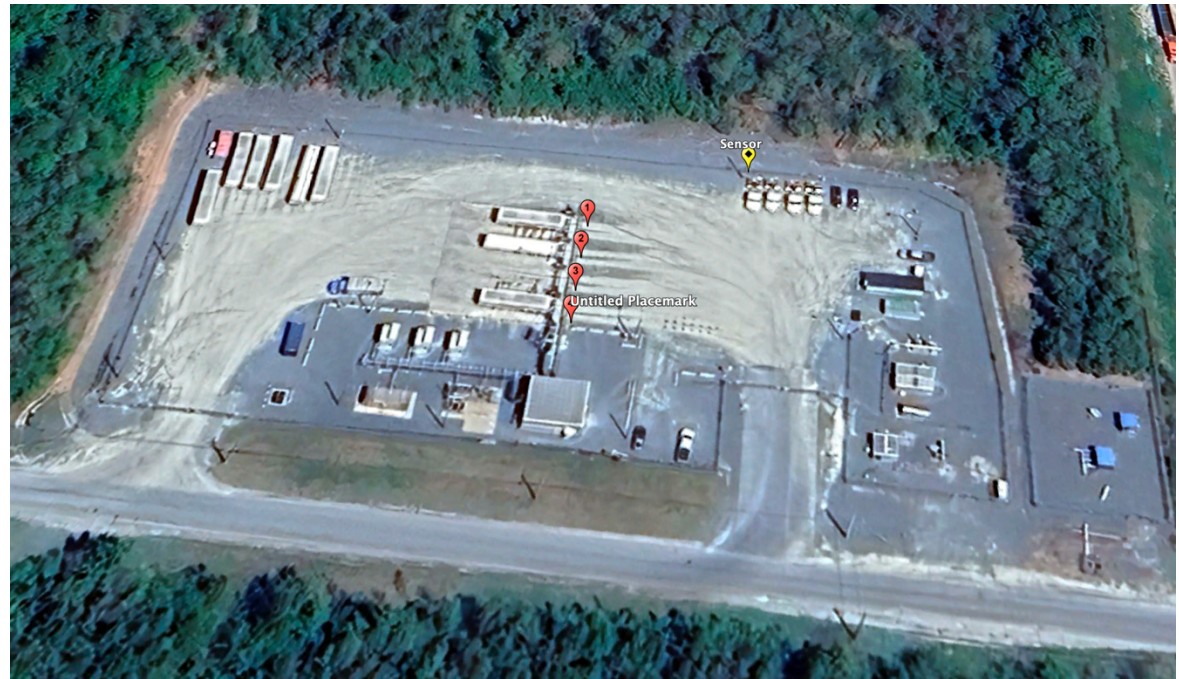

**Figure 5: Gas distribution site In Nova Scotia. Numbers 1-4 represent possible sources, and the yellow circle is the sensor's location.**
**(Image created by the authors using Google Earth. Map data: © 2025 Airbus, © Google.)**

## 3 Results

### 3.1 Sensor Placement Configuration for Synthetic Data

For the 456 synthetic scenarios, we calculated POD based on three categories: relative height, relative distance, and stability class of the air. To score true detections, TERRAFEX and the GI tool had to localize
at least one source within 10 meter to 100 meter distance. Figure 6 illustrates side-by-side plume dispersion of methane from two emitters placed 25 meters apart. The left-hand panel shows the Gaussian Dispersion Model (GDM) output result, the forward-simulated concentration field in parts per million by volume (ppmv), which is now tagged on the color bar. The highest concentrations (red areas) are immediately downwind of the sources, gradually decreasing (green to blue) as the plumes spread out and
dilute with distance. The black vertical line marks where the measurement transect is, where the two plumes have already merged to produce a single broad concentration footprint.

The right-hand panel is the methane concentration map reconstructed from TERRAFEX for the same case. Similarly, the color scale represents methane concentration in ppmv as the figure is labeled. The
combined plume pattern from the reconstructed field along the line of measurement is presented, with contour lines further defining the concentration gradient. From this comparison, it is evident that at the measurement point, the two plumes cannot be separated due to their overlap and may therefore be mistaken for a single source. This emphasizes the importance of measurement positioning if we had



measurements been taken closer to the sources, the two plumes would be more distinct, and the ability to
resolve individual sources would be improved.

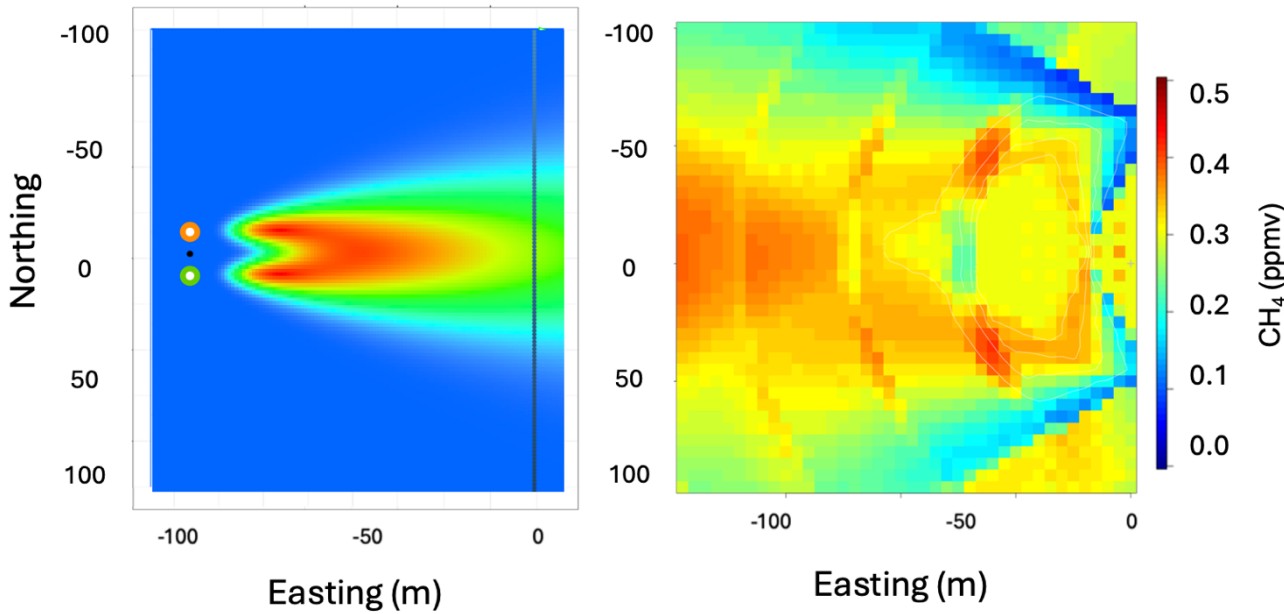

**Figure 6: Left) GDM forward model for two plumes within a well-pad, The line at x=0 shows the measurement point downwind the sources.; Right) TERREFAX map for the same scenario of two sources.**

Table 1 shows the POD calculation based on stability class for two sources with an emission rate of 41 $m^3$ $day^{-1}$ for each source for all 456 synthetic scenarios. As it can be observed, stability class D generally had the lowest localization accuracy. Still, configurations two and three, where at least one of the sources was not aligned with the sensor, also resulted in a smaller accuracy compared to when the sources were aligned with the sensor. Stability C drove the highest LA among all classes, but the POD was similar
within stability classes A, B, and C.

**Table 1: Uncertainty analysis of source detection performance across atmospheric stability classes for configurations for all scenarios in Fig 1.**

| Stability class | POD | FPF | FNF | LA |
|---|---|---|---|---|
| A_1 | 1.00 | 0.28 | 0.00 | 0.72 |
| A_2 | 0.69 | 0.68 | 0.13 | 0.32 |
| A_3 | 1.00 | 0.78 | 0.00 | 0.22 |
| B_1 | 0.95 | 0.32 | 0.03 | 0.68 |
| B_2 | 0.58 | 0.74 | 0.16 | 0.26 |
| B_3 | 1.00 | 0.78 | 0.00 | 0.22 |
| C_1 | 1.00 | 0.22 | 0.00 | 0.78 |





| | | | | |
|---|---|---|---|---|
| **C_2** | 0.67 | 0.79 | 0.09 | 0.21 |
| **C_3** | 1.00 | 0.75 | 0.00 | 0.25 |
| **D_1** | 0.89 | 0.38 | 0.07 | 0.62 |
| **D_2** | 0.40 | 0.86 | 0.18 | 0.14 |
| **D_3** | 0.63 | 0.82 | 0.10 | 0.18 |

The POD result is shown in  m.

Table 2 for the varying distance between the source and the sensor. In general, the POD was exceeded 50% even when the sources were less than 25 meters away from the sensor, but considering the ratio of FPF, the LA was highest when the distance was between 25 and 75 m. The FPF was also trending
downward for increasingly elevated distances between 100 and 125 m.

**Table 2: Uncertainty analysis of source detection performance across for source detection when varying distance between sources and sensor**

| Distance between sensor and source (m) | POD | FPF | FNF | LA |
|---|---|---|---|---|
| ≤ 25 | 0.50 | 0.67 | 0.25 | 0.33 |
| ≤ 50 | 0.73 | 0.33 | 0.20 | 0.67 |
| ≤ 75 | 0.90 | 0.44 | 0.06 | 0.56 |
| ≤ 100 | 0.82 | 0.78 | 0.04 | 0.22 |
| ≤ 125 | 0.93 | 0.39 | 0.04 | 0.61 |
| 125< | 1.00 | 0.69 | 0.00 | 0.31 |

It could easily be established that at a distance of 75 m from the edge, the system could map several locations of sources. Further afield, its accuracy declined by several orders and failed to tell between sources. The more distant the source, the more the system treated both as one emission point since sources emerged relatively close to each other while the detection tool had insufficient resolution. This effectively allows the sources to be made to appear to combine or merge, in which case separating the sources as
independent points of emission is most improbable.

Table 3 shows that LA decreases when the height difference exceeds 1 m. Although the POD is still relatively high, the major contribution is mainly for FPF and not for TP-that is, anomalies are effectively detected, but there is an error in finding the exact locations of the sources. The phenomenon was clearer
for stability classes A and B, where estimated source locations showed up closer to the sensor compared to the reality. Even sources whose actual position was not close to the sensor or aligned with it, the system predicted locations closer to the sensor location. This suggests that under unstable atmospheric conditions, the model's estimations may be biased toward placing the source closer to the sensor due to how the



plume disperses. There was an increase in the LA trend when the height difference was 4 m. This may
suggest that the relative distance between sources and sensor was a bigger control after a certain height
difference.

**Table 3: Uncertainty analysis of source detection performance across for source detection when varying heights between sources and sensor**

| The height difference between sources and sensor | POD | FPF | FNF | LA |
|---|---|---|---|---|
| 0 | 0.85 | 0.62 | 0.06 | 0.38 |
| 1 | 0.86 | 0.66 | 0.05 | 0.34 |
| 2 | 0.84 | 0.72 | 0.05 | 0.28 |
| 3 | 0.80 | 0.79 | 0.05 | 0.21 |
| 4 | 1.00 | 0.73 | 0.00 | 0.27 |
| 5 | 0.67 | 0.74 | 0.12 | 0.26 |

Figure 7 represents POD graphs for a) stability class, b) relative distance, and c) relative height between
source and sensor. The shaded area is a confidence interval (CI) starting from 50% at the top (dark shade)
to 95% (light shade). It is evident from a) that stability A, B, and C all resulted in high POD while used
for scenarios where at least one of the sources was aligned with the sensor. POD exceeded 50% for all
cases, but POD of 90% was only achieved for stability classes A, B, and C for cases when there was at
least one aligned source.
Filled areas are the region between the 50% confidence interval (CI) limit (top, dark) and the 95% CI
(bottom, light), indicating the uncertainty in POD estimation as a function of varying atmospheric and
source-sensor geometries. Red dots are the observed POD for the corresponding conditions, and the blue
line is the median modelled POD.

In panel (a), POD increases with the level of atmospheric instability, and the highest detection
probabilities are provided by stability classes A, B, and C. Specifically, instances where at least one of the
sources directly faced the sensor under these unstable-to-neutral conditions reported PODs over 90%. In
contrast, more stable (D-class and below) conditions presented lower PODs but still above
50% in all instances.

Panel (b) shows the effect of relative distance between source and sensor on POD. Interestingly,
POD increased with distance up to 125 meters, suggesting plume merging effects and
sensor location relative to combined footprint. Beyond this distance, POD remained high, approaching
100%. This reaction suggests that at certain downwind distances, plume overlap maximizes
detectability, and very close distances may limit detection due to thin plumes or misalignment.

Panel (c) presents the influence of relative height difference between source
and sensor. With increasing relative sensor height above the source, so did the POD,



with optimum values at maximum relative heights employed (up to 6 meters above the source).
This implies that raising the sensor enhances detection coverage by intersecting the plume more efficiently, particularly in unstable conditions when plumes increase more vigorously.

In general, all the graphs illustrate the complex interplay among atmospheric stability, alignment between source and sensor, and detection efficiency. The large confidence intervals in the more stable or 430 misaligned cases indicate the increased variability and uncertainty of POD estimation for such cases.



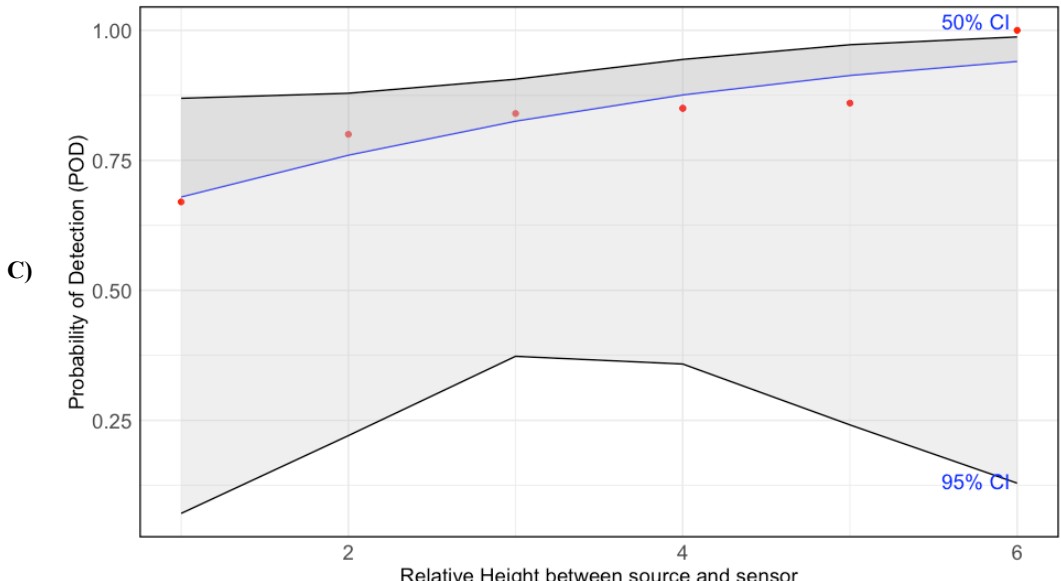

**Figure 7: Example of relative location between sources and a sensor. The red marks show the source locations found by the GI tool. The bottom line shows the 95% confidence interval and the top line represents 50% confidence interval.**

While POD is still above 50% for relative source location, it improves by increasing the distance between
the source and sensor. From c), it can be seen that POD is always high, but it improves for cases where the relative distance is below 3 m and then drops down for distances above 4 m.

In some of our measurements, the sensor was positioned more than 3 meters below the sources, making accurate localization nearly impossible. Although this was not an expected scenario for the application of TERRAFEX, this could happen in many cases and here we wanted to see the validity of results when
cases like this would happen. The base assumption in this study was an unknown source location, therefore controlling the height would not be possible. This was reflected in our synthesized experiment, where accuracy dropped off significantly with height difference. Table 3 reports the values, with LA dropping down from 0.38 down to 0.21 with the height difference reaching 3 meters, being in good agreement with the decreased accuracy found using Jacob et al. (2022), whom sources that place higher
or lower had worse localization.

## 3.2 Source Localization at Real Gas Distribution Site

Figure 8 illustrates the results for the gas collection site case study at the 4.5-meter height. The detected sources were within 10 meters of their actual locations, demonstrating that adherence to relative height and distance guidelines, derived from synthetic data, significantly enhances both POD and LA, potentially
reaching 100%. It should be noted that the measurement was conducted as a blind test, and information regarding the exact emission rates and their distribution across source locations was not disclosed. While



multiple sources appeared active based on plume simulations, no assumptions were made about their individual rates or timing.

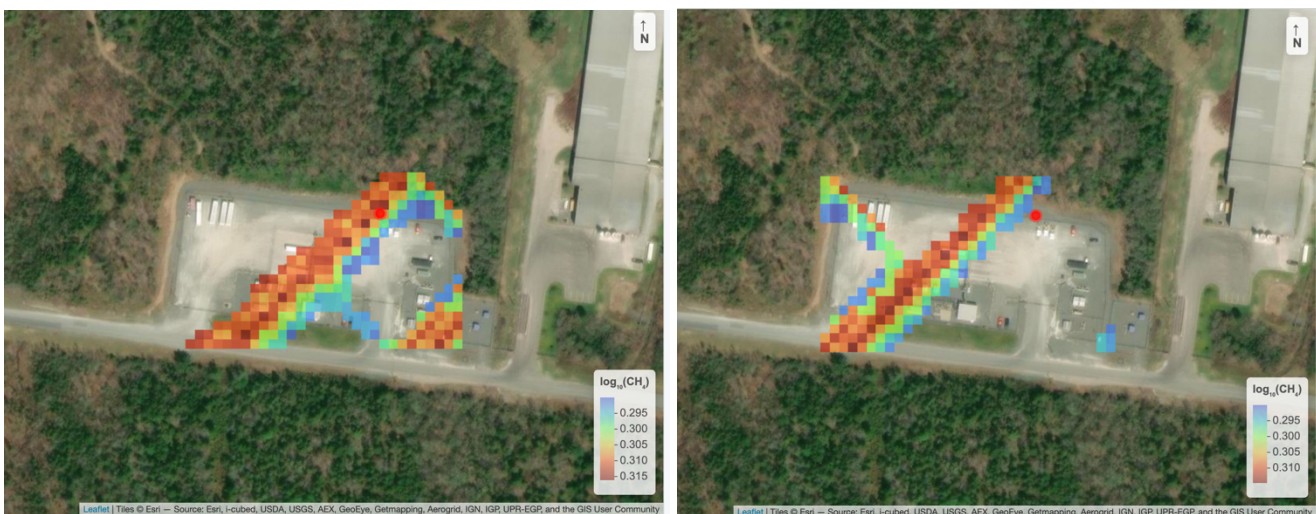


**Figure 8: Localization results from the TERRAFEX Gradient Indicator model for stability classes C (right) and D (left) for one day of measurement. The overlaid raster shows the $\log_{10}$-scaled estimated source regions, with warmer colors indicating higher likelihood, and the red circle indicating the sensor location.**

In this case study, adjusting the sensor height resulted in detecting the anomalies from sources and, therefore, making it possible to localize sources. Our results showed that following height and distance recommendations based on synthetic simulations led to accurate emission localization within 10 m of the actual source. Other real-world field studies also agree with these results, such as those conducted in the METEC facility. Bell et al. (2022), who worked on the CEMS at the METEC, also demonstrated 465    improvements in accuracy where sensor adjustments took into account differences in height, and detection ranges were similar to what is demonstrated here.

## 4 Discussion

This study applies the TERRAFEX model to estimate POD and LA for selected scenarios based on classes 470    of atmospheric stability, sensor-source elevation, and distance. Our research extends the current literature on CEMS by pointing out essential issues and opportunities with sensor-based methane detection and localization.

Our results agree with other studies on how air stability affects the accuracy of source localization (Chen et al., 2022; Williams et al., 2021). Neutral stability class (D) resulted in a lower level of accuracy in 475    localization (LA = 0.62 in D_1 from Table 1), as increased dispersion in air reduces the strength of the GI instrument. On the other hand, unstable atmospheric conditions (classes A, B, and C) enhanced POD and LA, thus supporting the conclusions of Ravikumar et al. (2019) regarding the effect of stability on the detection accuracy. This is in agreement with other studies by Daniels et al. (2024) and Cardoso-





Saldaña (2023), which have shown that air stratification has a direct effect on the performance of
continuous monitoring systems in oil and gas facilities.

One crucial determinant of detection efficacy was proximity of the sensor to the source. Our analysis
revealed that the POD was over 0.70 at distances less than 50 m; thus, results are in concurrence with
findings of Daniels et al. (2022) and Johnson et al. (2023) that short-range methods have significant
improved detection accuracy. Beyond 75 m, the accuracy of localization decreases because of increased
plume dispersion; LA = 0.56 at ≤75 m, in line with Thorpe et al. (2017). These results point to a serious
weakness in the CEMS network, as a similar inefficiency in detection was noted by Chen et al. (2023a)
in continuous monitoring studies. Because infrastructure related to the deployment of sensors has become
very minimal in oil and gas sectors, sensor location methodology has turned out to be the major
contributing factor for better detection reliability.

Our analysis of the sensor-source elevation discrepancies supports the results of Jacob et al. (2022) and
proves that LA decreases with height changes over 1 m, entailing an increased false positive rate. Cooper
et al. (2020) pointed out that sources of elevated or recessed emissions hinder the correct identification of
plumes. Ilonze et al. (2024) and Bell et al. (2023) examined the performance of various commercial
CEMS solutions in a similar setup and report a severe increase in false detections for complex setups.

From a legislative point of view, the detection threshold needs to be 90%. Our experiments show that this
detection threshold was routinely satisfied at 25 - 75 m distances in classes A, B, and C. That means
sensor-based methane detection works best in the A, B, and C stability classes. In D stability class neutral
conditions, POD was substantially decreased (0.40 in D_2, Table 1). As Ravikumar et al. 2018 stated,
classes of atmospheric stability need to be considered when designing the regulatory monitoring system.

Although our study provides important information, some limitations have to be considered. We used a
constant release rate of 41 m³/day with very low noise levels and could therefore not investigate effects
of signal-to-noise ratios on detection efficiency. Chen et al. (2024) and Daniels et al. (2023) report as
follows: The challenges are low when the signal-to-noise ratio is high; hence, high resolution is needed
to estimate small plumes at reasonable accuracy when the signal-to-noise ratio is low for $CH_4$ sensors.

According to Bell et al. (2024), such extreme variability within the signal-to-noise should be taken into
consideration by other studies for better capture in the variability of the emissions. Real complications
also arise outside the system, which can affect the detection performance. For instance, noise from
instruments, changing wind patterns, and localized baselines of methane might introduce uncertain
margins into gradient calculations and, thus drive up the rate of false alarms or even missed detections
(Jia et al., 2023). Besides, sensor placement strategies should consider changing infrastructure
configurations and high-density emission zones that will affect the concentration gradients and further
lower the localization accuracy. Daniels et al. (2024) and Wang et al. (2022) report that optimal sensor
placement is a huge challenge in continuous monitoring applications, especially for large oil and gas
facilities.

The expanding use of methane mitigation strategies involving CEMS-based solutions might be considered
another potential research avenue in the exploration of next-generation multi-sensor integrations based
on AI-driven atmospheric modeling. Adaptive machine learning techniques should incorporate methane
monitoring systems to enhance real-time detection precision, especially in volatile conditions. According
to Daniels et al. (2023), the ground-based CEMS data integrated with satellite and airborne measurements
provide a multiscale approach for comprehensive methane emission monitoring. Despite the latest



developments in continuous monitoring, performance remains critically influenced by meteorological variables to the sensor placing methodology. Herein, some suggestions are noise filtering methodologies, updated computation models, and strategic sensor placement for efficient detection of $CH_4$ that ensures compliance adherence. These limits show a perspective on how this research can increase the performance

of methane monitoring tools for more feasible emission control operations in the petroleum industry.

## 5 Conclusion

The objective of this study was to improve the localization of $CH_4$ emission sources in oil and gas facilities by adjusting sensor configuration settings and using more sophisticated analytical tools. We determined

the method's sensitivity in localizing the sources by changing relative source and sensor location and height by generating and analyzing synthetic data with the help of the Gaussian Dispersion Model and a Lagrangian Stochastic Back Trajectory Model (TERRAFEX). We used a Gradient Indicator (GI) tool to find the source locations accurately.

The results from our synthetic data analysis indicate that atmospheric stability, relative sensor-source

height, and distance are critical factors influencing the performance of the TERRAFEX model and GI tool in localizing emission sources. Our study consistently met this threshold for distances between 25 and 75 m under stability classes A, B, and C. However, stability class D significantly drove lower accuracy with generally lower LA and POD, thus pointing out the challenges in neutral atmospheric environments.

Second, localization accuracy decreased with the increase in the height difference between sensor and source; above 1 m, an evident increase of false positives shows this. Conclusively, this indicates small height differences of sensors with regard to sources provide more accurate locations, which falls in line with prior work.

The confidence of localization was sensibly assessed through POD, FPF and the  LA among 456 synthetic

situations. The performance of target localization was very different, and the LA was from 14% at the worse cases  to 78% at the better cases. The values of POP  ranged from 0.4 to 1.0 in different stabilities, and FPF was as high as 86% when the sensor-source alignment was poor or the elevation difference exceeded 3 m, indicating that even though the system works well at times, uncertainty of the results cannot be ignored. These numbers establish a strong quantitative foundation for consideration  of error and trade-

offs in practical applications.

While our fixed emission rate limited exploration of varying plume intensities, literature suggests that higher emission rates would improve localization accuracy, especially under unstable conditions.

While the approach performed well and put up a promising performance, there are also a few challenges that exist while applying it in real situations: instrumental noise, variations in the ambient environment,

operational constraints on control of relative heights and distances of sensors and sources among others can bring down LA and POD. Second, atmospheric stability conditions impose further restrictions: A, B, and C-conditions in practice make this less effective under truly neutral conditions, as dispersion dynamics interfere with source placement accuracy. This result is perfect for the controlled environments in general and monitoring by regulations of oil and gas facilities where sensor positioning can always be

chosen. Its adaptability to various emission rates and its possible integration with advanced real-time monitoring systems make it a valuable tool with which methane mitigation strategies can be substantially enhanced. However, extending this to a broader and more unpredictable scenario further requires honing




to reduce some challenges provided by noise, complicated terrains, and fluctuating environmental conditions.


## 6 Code and Data availability

All data processing, analysis, and visualization in this study were performed using R (version 2023.06.0). The complete R scripts for forward GDM model, and Gradient indicator tool, along with the created datasets and outputs of TERRAFEX supporting the findings of this study—including simulated plume 570 fields, detection probability outputs, and visualization files—are openly available at https://doi.org/10.5683/SP3/HPMOC7.

Both code and data are provided under the Creative Commons Attribution-NonCommercial-NoDerivatives 4.0 International (CC BY-NC-ND 4.0) license. Users are encouraged to cite the provided DOI when reusing these resources.


## 7 Author contribution

- Contributed to conception and design: A.K., M.G., N.N., and D.R.

- Contributed to data acquisition: A.K., N.N.

- Contributed to analysis and interpretation of data: A. K and M. G.

• Drafted and/or revised the article: A. K, M. G, N. N, and D. R.

- Final version revision: A. K, M. G, N. N, and D. R.

## 8 Competing interests

The authors declare that they have no conflict of interest.

## 9 Acknowledgement


We acknowledge the use of ChatGPT (OpenAI) to refine the text of this manuscript. The AI tool was used for language improvement and clarity enhancement, without influencing the scientific content or interpretation.



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
