# Peer review of "Optimizing Methane Emission Source Localization in Oil and Gas Facilities Using Lagrangian Stochastic Models and Gradient-Based Detection Tools"

_EGUsphere, 2025_

## Referee Comment (RC1)

**Review of *"Optimizing Methane Emission Source Localization in Oil and Gas Facilities Using Lagrangian Stochastic Models and Gradient-Based Detection Tools"***

The manuscript addresses a critical gap in methane emission monitoring: while detecting emissions is essential, *localizing their sources* is paramount for effective mitigation. The authors present a novel integration of the **TERRAFEX** Lagrangian stochastic model with a **Gradient Indicator (GI) tool**. This work have put large amount of efforts and defining several scenarios to try to advances continuous emission monitoring systems (CEMS) by improving spatial attribution under complex field conditions. Below, I offer general and specific feedback to strengthen the study's impact.
* * *
**General comments**
* * *
1- I am not fully convinced that this approach can be applied in a real world condition. Probably this can be further improved in the manuscript or explained in a better way. While the authors attempt to study this important topic and provide a new approach, they can possibly try to show how this approach is a good way to be applied in a real world-conditions. Otherwise, I would recommend that the authors the parameters they studied which influence the POD and/or LA.

2- Probably the authors can explain in the manuscript if the use of TERRAFEX can be also used for a site with more than one emitting source.

3- The presentation of figures can be improved (see comments below, mainly regarding Fig. 7) and some of formulas can be possibly re-defined (explained below mainly related to LA).

4- The manuscript is written in a good way, however for the improvement I have identified some editorial comments which can be considered.
* * *
**Detailed comments**
* * *
L45 :47 – is this underreporting for Canada or worldwide? In some cases the underreporting is higher than 1.5.

L187-188 – Rephrase, it is a bit vague.

L208 – How did you define the stability classes? Please add few words accordingly.

L230 – the 45° angle changes when the sensor placement increases from the first position as stated in L250. Or did you consider the 45° angle for all sensor locations?

L285 – why did you use Monin-Obukhov length instead of stability class?

Figure 4 – If the edge of well pad is 100 m away from the source, and the sensor position starts from the edge of the well pad at 10 m increment, then the source and sensor cannot be relatively as close as 10 meter to each other, right? See L282.

L315 – The FN was described before.

Figure 6 – Visually speaking, it seems that the plume dispersion from these two sources follow two different wind field. As you can see the plume originated from the north source tends to curve southward and vice versa for the south source. Can you please clarify?

L348 – Wouldn't you get two locations anyway from the TERRAFEX as it seems it mirrors the plume? It is only about the localization? Can you also do emission back trajectory for more than two locations using the same method?

Table 1, Table 2, Table 3 and L320 – It seems that the formula is for the FNF is not correct. The FNF is usually calculated using $n_{FN} / (n_{FN} + n_{TP})$. This has influence on the values in Table 1.

L321 – shouldn't be LA defined as $(n_{TP} + n_{TN}) / n_c$ or if you are focused on the emitting sources, shouldn't be the formula defined as $n_{TP} / n_c$? I would suggest to change the formula of LA to average detected distance to the true source +/- uncertainty (e.g. 1 standard deviation). For example something like this:

$$LA = \sqrt{(x_d - x_t)^2 + (y_d - y_t)^2}$$

In which $(x,y)_d$ is the location of detected source and $(x,y)_t$ is the true location of source. Then you can calculate the standard deviation from all the distances calculated.

Table 1, 2 and 3 – the sum of POD and FNF should be 1 following the abovementioned comment.

Figure 7 – I would recommend to change representation of the POD vs parameters and lines of CIs. Probably use POD as y axis and parameters as X-axis and show the 50% CI around the mean or median in the figures. On another point, I can see from Table 2 that POD for <100 values is lower than POD for <75 and <125 while in Figure 7 panel b this is not the case. Check the values.

Figure 8 – So it seems that the sources can be anywhere on the red pixels. Please elaborate how TERRAFEX can be useful in real world conditions. And why did you use the logarithmic scale?

L447 – if the information about the exact location of the sources were not disclosed, how can you determine that the detected sources were within the 10 m distance of actual locations?
* * *
**Editorial comment**
* * *
L40 – add parentheses for the year 2023, check referencing style. Also in L67 and L70.

L42 – Add reference to this after '…missions by 30% before 2030.'

L43 – Add reference to the contribution of O&G.

L103 and L105 and elsewhere– check the italic format of the reference.

L196 – check the subscript.

L205 – GDM needs to be spelled out here instead of Sect. 3.1.

L456 – POD?

---

## Referee Comment (RC2)

**Review of *"Optimizing Methane Emission Source Localization in Oil and Gas Facilities Using Lagrangian Stochastic Models and Gradient-Based Detection Tools"***

The manuscript addresses a critical gap in methane emission monitoring: while detecting emissions is essential, *localizing their sources* is paramount for effective mitigation. The authors present a novel integration of the **TERRAFEX** Lagrangian stochastic model with a **Gradient Indicator (GI) tool**. This work have put large amount of efforts and defining several scenarios to try to advances continuous emission monitoring systems (CEMS) by improving spatial attribution under complex field conditions. Below, I offer general and specific feedback to strengthen the study's impact.
* * *
*General comments*
* * *
1- I am not fully convinced that this approach can be applied in a real world condition. Probably this can be further improved in the manuscript or explained in a better way. While the authors attempt to study this important topic and provide a new approach, they can possibly try to show how this approach is a good way to be applied in a real world-conditions. Otherwise, I would recommend that the authors focus on the parameters they studied in the manuscript which influence the POD and/or LA.

2- Probably the authors can explain in the manuscript if the use of TERRAFEX can be also used for a site with more than two emitting sources.

3- The presentation of figures could be enhanced (particularly Figure 7, as detailed in the comments below). Additionally, some formulas may benefit from redefinition or clarification, especially those related to the LA approach (specific suggestions provided below).

4- The manuscript is well-written, but I have identified several editorial suggestions for further improvement.

5- As a recommendation for potential inclusion in the manuscript, please consider evaluating the applicability of the Other Test Method (OTM) 33A quantification method. This approach, developed by the EPA, is designed for stationary measurements of ambient methane emissions (mixing ratio or widely used term

concentration in industry) alongside simultaneous wind direction data. If feasible, you may explore integrating OTM 33A into your algorithm after completing source localization and distance determination. For reference, see: Korben et al. (2022), Omara et al. (2018), and EPA (2014)
* * *
*Detailed comments*
* * *
L45 :47 – is this underreporting for Canada or worldwide? In some cases the underreporting is higher than 1.5.

L187-188 – Rephrase, it is a bit vague.

L208 – How did you define the stability classes? Please add few words accordingly.

L230 – the 45° angle changes when the sensor placement increases from the first position as stated in L250. Or did you consider the 45° angle for all sensor locations?

L285 – why did you use Monin-Obukhov length instead of stability class?

Figure 4 – If the edge of well pad is 100 m away from the source, and the sensor position starts from the edge of the well pad at 10 m increment, then the source and sensor cannot be relatively as close as 10 meter to each other, right? See L282.

L315 – The FN was described before.

Figure 6 – Visually speaking, it seems that the plume dispersion from these two sources follow two different wind field. As you can see the plume originated from the north source tends to curve southward and vice versa for the south source. Can you please clarify?

L348 – Wouldn't you get two locations anyway from the TERRAFEX as it seems the algorithm mirrors plume? Can you also do emission back trajectory for more than two locations using the same method?

Table 1, Table 2, Table 3 and L320 – It seems that the formula is for the FNF is not correct. The FNF is usually calculated using $n_{FN} / (n_{FN} + n_{TP})$. This has influence on the values in Table 1.

L321 – shouldn't be LA defined as ($n_{TP}$ + $n_{TN}$) / $n_c$ or if you are focused on the emitting sources, shouldn't be the formula defined as $n_{TP}$ / $n_c$? I would suggest to change the formula of LA to average detected distance to the true source +/- uncertainty (e.g. 1 standard deviation). For example something like this:

$$LA = \sqrt{(x_d - x_t)^2 + (y_d - y_t)^2}$$

In which $(x,y)_d$ is the location of detected source and $(x,y)_t$ is the true location of source. Then you can calculate the standard deviation from all the distances calculated.

Table 1, 2 and 3 – the sum of POD and FNF should be 1 following the abovementioned comment (see comment related to Table 1, Table 2, Table 3 and L320).

Figure 7 – I would recommend to change representation of the POD vs parameters and lines of CIs. Probably it would be better to use POD as y axis and parameters as X-axis and show the 50% CI around the mean or median in the figures. On another point, I can see from Table 2 that POD for <100 values is lower than POD for <75 and <125 while in Figure 7 panel b this is not the case. Check the values.

Figure 8 – So it seems that the sources can be anywhere on the red pixels. Please elaborate how TERRAFEX can be useful in real world conditions. And why did you use the logarithmic scale?

L447 – if the information about the exact location of the sources were not disclosed, how can you determine that the detected sources were within the 10 m distance of actual locations?
* * *
*Editorial comment*
* * *
L40 – add parentheses for the year 2023, check referencing style. Also in L67 and L70.

L42 – Add reference to this after '…missions by 30% before 2030.'

L43 – Add reference to the contribution of O&G.

L103 and L105 and elsewhere– check the italic format of the reference.

L196 – check the subscript.

L205 – GDM needs to be spelled out here instead of Sect. 3.1.

L456 – POD?

References:

Korbeń, P., Jagoda, P., Maazallahi, H., Kammerer, J., Nęcki, J. M., Wietzel, J. B., Bartyzel, J., Radovici, A., Zavala-Araiza, D., Röckmann, T., and Schmidt, M.: Quantification of methane emission rate from oil and gas wells in Romania using ground-based measurement techniques, Elem. Sci. Anth., 10, 00070, https://doi.org/10.1525/elementa.2022.00070, 2022.

Omara, M, Zimmerman, N, Sullivan, MR, Li, X, Ellis, A, Cesa, R, Subramanian, R, Presto, AA, Robinson, AL.: Methane emissions from natural gas production sites in the United States: Data synthesis and national estimate. Environmental Science & Technology 52(21): 12915–12925. DOI: http://doi.org/10.1021/acs.est.8b03535, 2018.

EPA, Draft Other Test Method 33A: Geospatial Measurement of Air Pollution, Remote Emissions Quantification - Direct Assessment (GMAP-REQ-DA), Available from: https://www3.epa.gov/ttnemc01/prelim/otm33a.pdf, last access: 11 June 2025, 2014.

---

## Referee Comment (RC3)

**General comments**

This paper presents a new method for estimating methane emission source location on oil and gas sites using in situ point sensors. The method combines a backward Lagrangian dispersion model (TERRAFEX) with a gradient-based method for isolating enhancements in the resulting concentration maps. The authors use this method to evaluate detection and localization performance of the in situ sensors via a simulation study, where artificial sensor data is generated using a Gaussian dispersion model. I believe that the results of this simulation study are the primary contribution of this work, as they may be able to provide performance bounds that could be expected in practice. The authors also apply the method to real data collected on an oil and gas facility.

However, I have a number of major concerns that must be addressed before I am able to fully interpret the proposed method, the simulation study, and the case study using real data. My primary concern is that the methods are not described in sufficient detail to be fully understood. For example, the description of the Gradient Indicator tool is lacking, and as such, it's not clear to me how this method picks an estimated source location based on output from the backward Lagrangian simulation tool. This makes it hard to evaluate the accuracy metrics presented in the results. Additionally, it's not clear if the main result tables (Tables 1-3) and figure (Figure 7) are for one of the source-sensor configurations used in the simulation study, or an average over all source-sensor configurations. This makes the results hard to interpret, as different configurations could have very different detection / localization capabilities. There also appear to be inconsistencies between Table 3 and Figure 7(b). Finally, I think that the writing could be refined; some ideas are expressed imprecisely making it hard to follow at times. See below for a complete list of comments.

**Specific comments**
- L58-59: The authors should note that gaps in CEMS detections are a result of wind blowing between sensors and that these periods can be identified (and subsequently addressed) by using an atmospheric dispersion model. For example, see: https://doi.org/10.1021/acs.estlett.4c00687
- L60-64: The authors should note that there is variability in localization and quantification performance across the commercial solutions studied in Bell et al. 2022 (and in the subsequent ADED evaluations, see https://doi.org/10.1021/acs.est.3c08511 and https://doi.org/10.26434/chemrxiv-2024-f1znb-v2). Some commercial solutions struggle to localize emissions, as the authors note, but others perform this task relatively well when evaluated at METEC.
- L82-83: This is not entirely true, there are several open-source methods for CEMS inversions in the literature. See for example: https://doi.org/10.48550/arXiv.2506.03395, https://doi.org/10.1021/acsearthspacechem.2c00093, https://doi.org/10.1525/elementa.2023.00110, https://doi.org/10.5194/amt-11-

1565-2018. The authors are correct, however, that commercial CEMS solutions often do not make their algorithms completely open source.

- The concept of a footprint is discussed in the introduction and methods sections but is not fully defined. I recommend defining this term in the introduction given that it is a key idea in the manuscript.
- The concept of a "forward simulation" vs. a "backward simulation" should be defined in the introduction
- L145: "source weight function" is not defined.
- Second paragraph of 2.2. More detail about the GI method is necessary. I don't understand some key ideas from this paragraph (e.g., the x and y direction and the new vs. original matrices).
- Each box in Figure 1 needs more detail to be interpretable. I'm not sure what a lot of these boxes mean, even after reading the previous paragraphs in Section 2.2
- Section 2.2 would be aided by a Figure showing the concentration maps, and perhaps an example of a "hot spot"
- Figure 3 is not a "time series," as it is referred to on L271. A time series would plot the methane concentrations on the vertical axis with time on the horizontal axis. These plots show concentration as a function of wind direction.
- Fig 5: could there be other sources on this site besides the four you identified? If so, how would this impact the analysis? Some discussion of this point should be included.
- Fig 6: it's not clear which source-sensor configuration is being shown here. Are there many sensors along the x=0 line? Or just one at x=y=0?
- Fig 6: I'm not sure what the concentration field on the right is showing. I think more detail needs to be included in Section 2.1 when describing TERRAFEX, because it's not clear to me what the output from this model shows. Is it the estimated plume shape based solely on the sensor measurements? If so, why is the plume wider on the left and narrower on the right? Also, can the authors explain the other artifacts in this concentration field?
- Fig 6: It would be very useful to show where the GI method would identify the source given this concentration map.
- Table 1: Are these metrics averaged across all source-sensor configurations? If so, it would be very hard to interpret these numbers, as different arrangements may have very different detection characteristics.
- Figure 7: The 95% confidence interval almost spans the entire range of the POD parameter (0-1). Some discussion of this point should be included. How confident are you in these trends given the very large confidence intervals?
- Figure 7 Panel a: The stability classes are sorted so that the POD is increasing from low to high. This needs to be stated directly, otherwise this plot is misleading.
- Figure 7 Panel b: the 100 mark on the horizontal axis is not in the correct order. I think this is also plotted so that the POD is strictly increasing, which is misleading. The horizontal axis should be increasing distances.

- Figure 8: is one pixel selected as the source location estimate? Or is the entire red swath the source estimate? The red region covers the sources, but it also covers much of the rest of the site as well. How do you pick a source location from the concentration maps shown in this figure?
- The POD values in Tab 3 don't seem to line up with the values in Fig 7 panel C. This needs to be checked.

**Technical corrections**

- Localization accuracy (LA) acronym is defined twice in the abstract
- Methane is defined as "CH4" twice in the first paragraph of the introduction, but the authors continue to use "methane" throughout the paper.
- L53: Jia et al. 2023 has been published: https://doi.org/10.1038/s41598-025-99491-x
- L59: There is no Daniels et al. 2022 in the list of references. Do the authors mean to say Daniels et al. 2023 or Daniels et al. 2024?
- L71: The Gaussian puff and plume models studied in Jia et al. 2025 are not back-trajectory methods. They are forward models that simulate the transport of methane from the source to the sensor.
- L91-98: This discussion would be clearer if the authors first provide a clear definition of both "concentration footprint" and "flux footprint."
- L113: "localization" is defined here but used multiple times earlier in the introduction. It should be defined at its first use.
- L170: need to provide some information about your coordinate system before you say things like the "x and y directions."
- L205: GDM not defined
- Eq 1 is missing a plus sign
- Figure 2 needs a legend for the x's and the red dashed line.
- L297: a more precise definition of POD should be used. It looks like there is a repeated sentence here as well.
- Your definition of localization accuracy (LA) is more commonly referred to as "positive predictive value" or "precision." It might be better to use these more common phrases.
- In the FNF equation, it is not clear what the $n_c$ refers to. It would be better to write out exactly how $n_c$ is calculated (e.g., $n_{tp} + n_{fn}$), as is done with the other equations.

---

## Author Comment (AC1)

Preprint egusphere-2025-644

**Optimizing Methane Emission Source Localization in Oil and Gas Facilities Using Lagrangian Stochastic Models and Gradient-Based Detection Tools**

Afshan Khaleghi[1,2], Mathias Göckede[3], Nicholas Nickerson[4], David Risk[1]

[1] Department of Earth and Environmental Sciences St. Francis Xavier University Antigonish, Nova Scotia, Canada
[2] Department of Process Engineering, Memorial University of Newfoundland, Newfoundland and Labrador, Canada
[3] Max Planck Institute for Biogeochemistry, Jena, Germany
[4] Eosense Inc, Nova Scotia, Canada

*Correspondence to*: Afshan Khaleghi (akhaleghi@stfx.ca)

**Response to Reviewer (Dr. Hossein Maazallahi)**

| Review Comment | Author' Response | Line changed |
|---|---|---|
| **Editor** | | |
| I am not fully convinced that this approach can be applied in a real world condition. Probably this can be further improved in the manuscript or explained in a better way. While the authors attempt to study this important topic and provide a new approach, they can possibly try to show how this approach is a good way to be | Response: The case study provided in the manuscript is a real-world source localization scenario. We were blind to where the sources are. In this case study there were a total of 4 sources identified. To improve the visualization, the figure will be changed to show how the gradient indicator finds the sources closer to where they are located. | |

| | | |
|---|---|---|
| applied in a real world-conditions. Otherwise, I would recommend that the authors focus on the parameters they studied in the manuscript which influence the POD and/or LA. | | |
| Probably the authors can explain in the manuscript if the use of TERRAFEX can be also used for a site with more than two emitting sources. | Response: There were four potential fugitive emission sources, and CH4 was measured using an Axetris LGD Compact-A CH4 with 0.01 ppm precision at 2 Hz frequency | |
| The presentation of figures could be enhanced (particularly Figure 7, as detailed in the comments below). Additionally, some formulas may benefit from redefinition or clarification, especially those related to the LA approach (specific suggestions provided below). | This will be considered for the revised version. | |
| The manuscript is well-written, but I have identified several editorial suggestions for further improvement. | Response: Editorial improvements will be implemented in the revised version. | |
| As a recommendation | Thanks for the suggestion. This study was an attempt to merge with the OTM33A concept. In other words, the attempt is to | |

| | | |
|---|---|---|
| for potential inclusion in the manuscript, please consider evaluating the applicability of the Other Test Method (OTM) 33A quantification method. This approach, developed by the EPA, is designed for stationary measurements of ambient methane emissions (mixing ratio or widely used term concentration in industry) alongside simultaneous wind direction data. If feasible, you may explore integrating OTM 33A into your algorithm after completing source localization and distance determination. For reference, see: Korben et al. (2022), Omara et al. (2018), and EPA (2014). | localize the sources to make it possible to use OTM for cases where source locations are unknown. | |
| L45 :47 – is this underreporting for Canada or worldwide? In some cases the underreporting is higher than 1.5 | Response: The studies mentioned in the paper for 1.5 times underreporting are all in Canada. | |

| | | |
|---|---|---|
| L187-188 – Rephrase, it is a bit vague. | We propose to make this more straightforward with: "It's important to note that a gradient length indicator can, at best, provide an approximate estimate of the source location." | |
| L208 – How did you define the stability classes? Please add few words accordingly. | Response: For synthetic data, the stability classes were chosen to vary from A to D, and the sigma values are calculated using Turner 1970 as described in the paper. This can be added to the case study: "The stability class for each measurement day was defined using data from the closest airport." | |
| L203 –the 45 angle changes when the sensor placement increases from the first position, as stated in L250. Or did you consider the 45-degree angle for all sensor locations? | Suggested modification: "The alignment angle is always relative to the line that passes (0,0) over the edge of the well-pad." | |
| L285 – Why did you use Monin-Obukhov length instead of stability class? | The Lagrangian method uses Monin-Obukhov length as described in lines 142-147 | |
| Figure 4 – If the edge of well pad is 100 m away from the source, and the sensor position starts from the edge of the well pad at 10 m increment, then the source and sensor cannot be relatively as close as 10 meter to each other, right? See L282. | The edge of the well-pad is located at (0,0), as shown in Figure 2. So when the sensor is located at (0,0), the source at (-10, 0) is 10 meters away from the sensor. That is the case shown in Figure 2, first scenario. | |

| | | |
|---|---|---|
| | Suggested modification to Figure 4:

[Figure]
 | |
| L315 – The FN was described before. | This extra text can be removed. | |
| Figure 6 – Visually speaking, it seems that the plume dispersion from these two sources follow two different wind field. As you can see the plume originated from the north source tends to curve southward and vice versa for the south source. Can you please clarify? | There is a difference between these two, as described before. One is the Lagrangian back trajectory that shows the sum of backtracked concentrations from the sensor. For the right picture, it can be seen that the sum is more intense around the (-100,0), which is the superposition of the two plumes. For clarification, the gradient indicator results can be added: | |

[Figure]

| | |
|---|---|
| L348 – Wouldn't you get two locations from the TERRAFEX, as it seems the algorithm mirrors plume? Can you also do an emission back trajectory for more than two locations using the same method? | Response: High-quality areas will appear when two sources are close, making it harder to pinpoint the exact location. The second point is an excellent observation, which is addressed in the case study involving four sources under real-world conditions. |
| Table 1, Table 2, Table 3 and L320 – It seems that the formula is for the FNF is not correct. The FNF is usually calculated using $n_{FN}$ / ($n_{FN}$ + $n_{TP}$). This has influence on the values in Table 1. | The reviewer is correct that Equation 3 was in error. The FNF value should be corrected in Table 1, Table 2, and Table 3. |

| | | |
|---|---|---|
| L321 – shouldn't be LA defined as $(n_{TP} + n_{TN}) / n_c$, or if you are focused on the emitting sources, shouldn't be the formula defined as $n_{TP} / n_c$? I would suggest to change the formula of LA to average detected distance to the true source +/- uncertainty (e.g. 1 standard deviation). For example something like this: LA = $\sqrt{(x_d - x_t)_2 + (y_d - y_t)_2}$

In which $(x,y)_d$ is the location of detected source and $(x,y)_t$ is the true location of source. Then you can calculate the standard deviation from all the distances calculated. | Response: We are looking at "How many detections were correct," which only concerns TP and FP as described in Eq. 5. $(n_{TP} + n_{TN}) / n_c$

It is defined as overall correctness. Our main goal was to control the fraction of corrected detections. We wanted to be able to define TP and FPs here, so we decided to assume that if a source is 10 m away from its actual location, it is still valid as a detection. | |
| Table 1, 2, and 3 – the sum of POD and FNF should be 1 following the abovementioned comment (see comment related to Table 1, Table 2, Table 3 and L320). | Response: FNF was corrected in a comment above. | |

| | | |
|---|---|---|
| Figure 7 – I would recommend to change representation of the POD vs parameters and lines of CIs. Probably it would be better to use POD as y axis and parameters as X-axis and show the 50% CI around the mean or median in the figures. On another point, I can see from Table 2 that POD for <100 values is lower than POD for <75 and <125 while in Figure 7 panel b this is not the case. Check the values. | I am not sure if I understand the difference here. POD is the y-axis, and the parameter is on the x-axis. Also in Figure 7, b <100 comes before 75 and 125.

Response: The graph from Table 2 should be adjusted to display the x-axis in increasing order. | |
| Figure 8 – So it seems that the sources can be anywhere on the red pixels. Please elaborate how TERRAFEX can be useful in real world conditions. And why did you use the logarithmic scale? | Response: Thanks for pointing this out. We propose replacing the figure with a clearer one to show the gradient indicator localization.
The log scale is applied purely for visualization to better distinguish low background values without affecting the underlying data. | |
| L447 – if the information about the exact location of the sources were not disclosed, how can you determine that the detected sources were within the 10 m distance of actual locations? | Thank you for the comment. We acknowledge that the wording may have led to confusion. To clarify, the magnitudes of the sources were not disclosed, but the locations of potential sources were known to the team (as shown in Figure 5). We have revised the manuscript to make this distinction more straightforward and avoid similar misunderstandings.
Suggested modification in text: "Although the source magnitudes were not disclosed during the experiment, the approximate locations of the emission sources were known to | |

| | the research team, as shown in Figure 5. This allowed us to assess detection accuracy based on proximity to the known locations." | |
|---|---|---|
| L40 – add parentheses for the year 2023, check referencing style. Also in L67 and L70. | Should be repaired. | |
| L42– Add reference to this after '…misdions by 30% before 2030.' | Should be repaired. | |
| L43 – Add reference to the contribution of O&G. | Should be repaired. | |
| L103 and L105 and elsewhere– check the italic format of the reference. | Should be repaired. | |
| L196 – check the subscript. | Should be repaired. | |
| L205 – GDM needs to be spelled out here instead of Sect. 3.1. | Should be repaired. | |
| L456 – POD? | ? | |

---

## Author Comment (AC2)

Preprint egusphere-2025-644

**Optimizing Methane Emission Source Localization in Oil and Gas Facilities Using Lagrangian Stochastic Models and Gradient-Based Detection Tools**

Afshan Khaleghi[1,2], Mathias Göckede[3], Nicholas Nickerson[4], David Risk[1]

[1] Department of Earth and Environmental Sciences St. Francis Xavier University Antigonish, Nova Scotia, Canada
[2] Department of Process Engineering, Memorial University of Newfoundland, Newfoundland and Labrador, Canada
[3] Max Planck Institute for Biogeochemistry, Jena, Germany
[4] Eosense Inc, Nova Scotia, Canada

*Correspondence to*: Afshan Khaleghi (akhaleghi@stfx.ca)

**Response to Reviewer (Anonymous Referee)**

| Review Comment | Author' Response | Line changed |
|---|---|---|
| **Editor** | | |
| For example, the description of the Gradient Indicator tool is lacking, and as such, it's not clear to me how this method picks an estimated source location based on output from the backward Lagrangian simulation tool. This makes it hard to evaluate the accuracy metrics presented in the results. | Thanks for pointing this out. I propose to add the text below to include more description on the Gradient Indicator tool: "Let $C(x,y)$ denote the $CH_4$ concentration matrix where $x,y \in \{1,2,\ldots, N\}$, and N is the number of grid cells in each spatial dimension. Each cell represents a grid cell with a specific resolution. The matrices are reoriented such that the origin corresponds to the sensor location and rows increase in the downwind direction. To identify spatial gradients consistent with plume structure under a dominant wind direction, a 1D moving window of length (L), varying in a loop starting from one, is applied along each row i. For each window starting at column index j, a gradient sub vector is $g_c$ and the window is flagged as a decreasing gradient if the rank order of $g_c$ Satisfies | |

| | | |
|---|---|---|
| | $$g_c = [C(x, y), C(x, y + 1), \dots, C(x, y + L - 1)]$$ $$rank(g_c) = (L, L - 1, \dots, 1)$$ The grid coordinates (x,y) are stored as valid gradient start points if the condition is met. As shown in **Error! Reference source not found.**, the gradient length will increase incrementally with each loop and proceed in both x and y directions until at least one of the top 5% highest concentration points remains on the map. At each step, the number of times each point persists across varying gradient lengths is recorded as a resistance rate. Points with the highest resistance rates—those consistently identified as strong gradients—are then filtered to ensure they fall within the dominant wind direction sector, confirming that they are positioned upwind of the sensor." | |
| Additionally, it's not clear if the main result tables (Tables 1-3) and figure (Figure 7) are for one of the source-sensor configurations used in the simulation study, or an average over all source-sensor configurations. This makes the results hard to interpret, as different configurations could have very different detection / localization capabilities. | Thank you for the helpful suggestion. To avoid confusion for future readers, we have revised the text in the Results section to indicate that Tables 1–3 and Figure 7 represent aggregate results across all source–sensor configurations, rather than a single case. We now explicitly state that distance-based groupings were used to interpret results across these varied configurations, while other variables (e.g., height difference and stability class) were examined in isolation. Suggested modification: "Thank you for the helpful suggestion. To avoid confusion for future readers, we have revised the text in the Results section to indicate that Tables 1–3 and Figure 7 represent | |

| | | |
|---|---|---|
| | aggregate results across all source–sensor configurations, rather than a single case. We now explicitly state that distance-based groupings were used to interpret results across these varied configurations, while other variables (e.g., height difference and stability class) were examined in isolation." | |
| There also appear to be inconsistencies between the Table 3 and Figure 7(b). | Table 3 is presented in Figure 7(c). There is an order issue in Figure 7 (b) that is suggested to be fixed as follows:  | |
| I think that the writing could be refined; some ideas are expressed imprecisely, making it hard to follow at times. | Thank you for this valuable feedback. To improve clarity and precision, we are undertaking a thorough manuscript revision. Specifically:

• We will **revise imprecise or ambiguous sentences** to communicate each idea clearly and concisely.
• We are having **independent colleagues with domain expertise** review the revised text to help identify areas where the language may remain unclear.

These efforts will help ensure the manuscript is more accessible to scientific readers and less prone to misinterpretation. | |

| | | |
|---|---|---|
| L58-59: The authors should note that gaps in CEMS detections result from wind blowing between sensors. These periods can be identified (and addressed) using an atmospheric dispersion model. For example, see: https://doi.org/10.1021/acs.estlett.4c00687 | Thanks for pointing this out. Our recent work on sensor placement optimization targets this limitation by showing that, with strategic placement, even a single sensor can effectively capture emissions from multiple directions and sources, thereby reducing the occurrence of non-detect periods. The paragraph was modified to point to this publication

"While combining multiple techniques can enhance source localization, stationary sensors still struggle to detect and differentiate emissions from multiple sources. Continuous Emission Monitoring Systems (CEMS), as noted by Jia et al. (2023), tend to mostly measure ambient methane concentrations rather than direct emission rates, thereby raising the issue of source identification when there are more than one emitters. Wang et al. (2022) stated that 1-minute average frequency for CEMS improves the chance of detecting 24-hourlived emissions that occur briefly. However, additional operator input is often necessary to refine accuracy. It is also important to note that detection gaps in CEMS can occur when wind advects emissions between sensors, resulting in periods where elevated concentrations go unrecorded. As demonstrated by Daniels et al. (2024), such "nondetect times" can be systematically identified and addressed using atmospheric dispersion modeling—specifically through the use of a probabilistic framework that reconstructs likely emission durations by simulating plume transport and wind-driven | |

| | | |
|---|---|---|
| | sensor coverage."
Daniels, W. S.; Jia, M.; Hammerling, D. M. Estimating Methane Emission Durations Using Continuous Monitoring Systems. *Environ. Sci. Technol. Lett.* **2024**, *11* (11), 1187–1192. https://doi.org/10.1021/acs.estlett.4c00687 | |
| L60-64: The authors should note that there is variability in localization and quantification performance across the commercial solutions studied in Bell et al. 2022 (and in the subsequent ADED evaluations, see https://doi.org/10.1021/acs.est.3c08511 and https://doi.org/10.26434/chemrxiv-2024-f1znb-v2). As the authors note, some commercial solutions struggle to localize emissions, but others perform this task relatively well when evaluated at METEC. | Thank you for pointing this out. We agree that the language should better reflect the variability across commercial CEMS solutions. Not all systems perform poorly—some demonstrate strong localization under ideal testing conditions.

Suggested text for manuscript: "Performance across commercial CEMS solutions varies considerably under controlled testing. While several point-sensor network systems often exhibited low localization accuracy ($< 50\%$) at the equipment unit level (Ilonze et al. 2024, Cheptonui et al. 2025), scanning/imaging solutions reliably achieved higher accuracy and precision ($> 50 - 90\%$) (Zimmerle et al. 2025, Cheptonui et al. 2025). Retesting of some solutions showed measurable improvements over time in localization and detection capabilities under METEC protocols (Day et al. 2024). Our analysis focuses specifically on stationary PSN configurations and the inherent limitations in source localization using downwind concentration measurements without supplementary modeling or imaging support." | |

| | | |
|---|---|---|
| L82-83: This is not entirely true; there are several open-source methods for CEMS inversions in the literature. See, for example: https://doi.org/10.48550/arXiv.2506.03395, https://doi.org/10.1021/acsearthspacechem.2c00093, https://doi.org/10.1525/elementa.2023.00110, https://doi.org/10.5194/amt-11-1565-2018. The authors are correct, however, that commercial CEMS solutions often do not make their algorithms completely open source. | Proposed revised text: "While several open-source inversion methods for CEMS data have been proposed in the academic literature (Liang et al., 2024; Barkley et al., 2022; Daniels et al., 2023; Fiehn et al., 2018), most commercially available CEMS solutions rely on proprietary algorithms—often involving some form of back-trajectory or dispersion modeling—which limits transparency and independent validation of source attribution."

• Daniels et al. 2025. https://doi.org/10.48550/arXiv.2506.03395
• Weidmann et al. 2022. https://doi.org/10.1021/acsearthspacechem.2c00093
• Daniels et al. 2023. https://doi.org/10.1525/elementa.2023.00110
• Aden et al. 2018 https://doi.org/10.5194/amt-11-1565-2018 | |
| The concept of a footprint is discussed in the introduction and methods sections, but is not fully defined. I recommend defining this term in the introduction, given that it is a key idea in the manuscript. | Thank you for emphasizing the importance of fully defining the concept of a footprint. We agree that it is a key term in the manuscript and that the current explanation could be more precise and comprehensive.

We have revised the introduction to include a more detailed definition to address this. The new text explains that a footprint quantifies the spatial sensitivity of a measurement to upwind surface emissions and distinguishes between concentration and flux | |

|  | footprints, including their respective units and applications. This revision clarifies the role of footprints in our study and reduces the risk of confusion for readers unfamiliar with the concept.

In atmospheric research, a footprint describes the source area from which air is arriving at a measuring point of an eye-friendly gas sensor. This is an illustration of how emissions from the multiple sources spread and be seen at the sensor, considering how atmospheric transport and dispersion work.

There are two common types of footprints:

• A concentration footprint describes the spatial sensitivity of a sensor to unit emissions at different locations and has units of seconds per square meter ($s\ m^{-2}$). This unit expresses how much a unit emission from a specific area ($1\ m^2$) contributes to the concentration at the sensor over time. The concentration footprint is especially relevant when interpreting measurements from point-in-space sensors or passive samplers that report mixing ratios (e.g., ppm or ppb).
• On the other hand, a flux footprint indicates the contribution of surface-atmosphere exchanges (e.g., methane fluxes) from different upwind areas to the total flux measured at |  |

| | | |
|---|---|---|
| | the sensor. It is expressed in square meters per square meter ($m^2\ m^{-2}$), effectively a dimensionless ratio that describes what fraction of the measured flux originates from each unit area. This is commonly used in eddy covariance (EC) studies or controlled release quantification.

Footprints are often a product of Lagrangian air parcel models or Gaussian atmospheric dispersion models, which simulate the movement of parcels of air over time and distance. Properties: The dimensions and shape of the footprint are governed by wind speed, turbulence, measurement height (expressed as z), atmospheric stability, and surface roughness. Under stable, low-wind conditions, high localization potential is observed through narrow and elongated footprints ($Czs > 0.1$) while broad footprints with $Czs < 0.01$ are associated to enhanced spatial uncertainty in source attribution driven by wind extremes and high turbulence.

The footprint models in this study are important to connect measured concentrations with potential upwind emissions sources, which improves source apportionment from monitoring sensors that have only a single exposure point. | |
| The concept of a "forward simulation" vs. a "backward simulation" should be defined in the introduction | This is a great point. We propose to add the following text manuscript: "In atmospheric transport modeling, a forward simulation predicts the downwind concentration field resulting from a | |

| | known or assumed emission source, often using meteorological data to simulate dispersion. In contrast, a backward simulation (also known as a back-trajectory or retro-plume approach) traces the path of air parcels in reverse to identify the potential upwind source regions contributing to observed concentrations at a sensor. These critical concepts support improved localization and quantification practices (Aubinet et al., 2012)." | |
|---|---|---|
| L145: "source weight function" is not defined. | That is a valid point. This has all been explained in Göckede et al.'s 2004-2006 studies.

"A source weight function describes how sensitive a measurement at the sensor is to potential emissions at different upwind locations over time. Conceptually, it quantifies the influence a unit emission from any given location would have on the observed concentration. That Implies a footprint but not actually a footprint This is the source weight function, which describes the mathematical relationship, while footprint refers to the map or spatial representation of that relationship for visualization and interpretation. In Lagrangian back-trajectory models, the source weight function is typically computed by tracing air parcels backward and assessing their time within each grid cell, adjusted by atmospheric parameters like turbulence and mixing height. For a detailed explanation of these concepts in the context of flux modeling, see Lin et al. (2003) and Kljun et al. (2015)." | |

| | | |
|---|---|---|
| Second paragraph of 2.2. More details about the GI method are necessary. I don't understand some key ideas from this paragraph (e.g., the x and y direction and the new vs. original matrices). | This is addressed in a comment above | |
| Each box in Figure 1 needs more detail to be interpretable. I'm not sure what a lot of these boxes mean, even after reading the previous paragraphs in Section 2.2 | This is addressed in a comment above that explained the GI method with formulation. Figure 1 can be replaced with this one

[Figure]

The text for section 2.2 can be modified as follows: "The main loop of the GI tool consists of a ranking method to flag the highest concentration points using different gradient lengths (cell numbers) starting from one. This is an assumption due to a lack of information from the source location (meaning that each concentration point is being flagged as high compared to itself). As shown in Figure 1, based on the mathematical concept of the GI method, length L will be selected starting from one, meaning that each (x, y) cell is maximum compared to itself in both x and y directions. The length will increase incrementally until at least one of the remaining points on the matrix surpasses the top 5% $CH_4$ concentration cells on the original matrix. The recurring gradient loop counts the persistence of points with different lengths and will only keep the high persistence points | |

| | from the TERRAFEX contribution source maps." | |
|---|---|---|
| Section 2.2 would be aided by a Figure showing the concentration maps, and perhaps an example of a "hot spot" | Text to be added: "An example of concentration maps and hot spots located by the GI tool is shown in Figure 6." | |
| Figure 3 is not a "time series," as it is referred to on L271. A time series would plot the methane concentrations on the vertical axis with time on the horizontal axis. These plots show concentration as a function of wind direction. | Correct. We propose this fix: "Methane concentrations vs wind direction at a sensor receptor, arriving from two points of emission upwind, driven by real varying winds representative of four different stability classes" | |
| Fig 5: Could there be other sources on this site besides the four you identified? If so, how would this impact the analysis? Some discussion of this point should be included. | Thank you for this thoughtful comment. You're correct that an overreliance on prior site knowledge could appear to contradict the potential of Lagrangian modeling to identify unknown emission sources.

Our analysis did not rely exclusively on prior knowledge of the four source locations. Instead, the spatial and temporal gradients in the sensor data—interpreted through the Lagrangian back-trajectory model—consistently pointed to four distinct emission regions, which aligned well with the known infrastructure. This agreement serves as mutual corroboration: site records confirmed these were plausible source locations, while the modeling supported their presence based on observed concentration patterns.

We believe that if additional, unaccounted-for sources had been present—especially of comparable magnitude—they would have produced distinct anomalies or | |

| | | |
|---|---|---|
| | unexplained gradients during measurement. For example, a hidden source south of the current sensor array would have likely shown elevated concentrations during southerly winds that could not be attributed to the known four sources. We did not observe any detectable unexplained patterns in the data.
Text to be added:
"Our analysis revealed four spatially distinct source regions following analysis of modeled upwind sensitivity patterns and concentration gradients observed in the measurements. This coincided with known site infrastructure and assured us that we were correct in our interpretation of the data. While we cannot rule out the existence of minor or intermittent unmonitored sources, we observed no anomalous gradients or unexplained patterns that would suggest significant unaccounted emissions. This suggests that both site inspection and modeling successfully captured the dominant sources." | |
| Fig 6: It's unclear which source-sensor configuration is shown here. Are there many sensors along the x=0 line? Or just one at x=y=0? | It is the second configuration. There is only one sensor at x=y=0 | |
| Fig 6: I'm not sure what the concentration field on the right is showing. I think more detail needs to be included in Section 2.1 when describing TERRAFEX, because it's unclear what the output from this model shows. Is it the estimated plume shape based solely on | We appreciate the reviewer's request for clarification. We have expanded Section 2.1 to describe better the TERRAFEX model, including how it reconstructs concentration fields using a back-trajectory framework. We also revised the caption of Figure 6 to explain that the right panel is not a | |

| | | |
|---|---|---|
| the sensor measurements? If so, why is the plume wider on the left and narrower on the right? Also, can the authors explain the other artifacts in this concentration field? | forward-simulated plume but a reconstructed field that represents the spatial influence of upwind sources on sensor measurements. The apparent asymmetries and shape artifacts are due to the underlying grid resolution, smoothing method, and the limited angular sensitivity of sensors under certain wind conditions.
The paragraph before the figure will be revised as follows:

"The right-hand panel shows the methane concentration field reconstructed using the TERRAFEX model for the same case. TERRAFEX uses back-trajectory simulations based on wind data to estimate the upwind contribution of potential source areas to each sensor reading. The resulting field is a probabilistic reconstruction of near-surface methane concentrations, derived by combining those back-trajectories with the observed $CH_4$ enhancements. The color scale represents methane concentration in ppmv, consistent with the left-hand panel. The reconstructed plume exhibits asymmetry and apparent artifacts—such as a wider spread on the upwind side and narrowing near the sensor line—due to the limited angular sensitivity of the sensors, interpolation over a discrete grid, and wind temporal variability during the reconstruction window.

This comparison demonstrates that at the downwind measurement location, the two | |

| | | |
|---|---|---|
| | plumes combine, making it difficult to distinguish the two plumes as separate sources. This illustrates the significance of sensor placement: at measurement locations that are downwind of the two source locations, if measurements had been taken at or near the source (the upwind part of the domain), the spatial separation of the two plumes would be maintained, creating good opportunities for localizing multiple sources." | |
| Fig 6: It would be very useful to show where the GI method would identify the source given this concentration map. | This figure can be added to show where the Gradient Indicator would identify the source.  | |
| Table 1: Are these metrics averaged across all source-sensor configurations? If so, it would be very hard to interpret these numbers, as diLerent arrangements may have very different detection characteristics. | This has already been discussed in a comment above | |
| Figure 7: The 95% confidence interval almost spans the entire range of the POD parameter (0-1). Some discussion of this point should be included. How confident are you in these trends given the very large confidence intervals? | Despite the wide CIs, the median (50%) trend remains robust, and in most cases, the POD exceeds 50%, even under less favorable conditions. That said, the uncertainty captured in the 95% CI highlights that individual detection events are more variable, and performance under these conditions should be interpreted | |

cautiously.

We propose to revise our description in the text:

"In panel a, POD increases with atmospheric instability, with the highest detection probabilities observed under stability classes A, B, and C. In these unstable-to-neutral conditions, detection rates often exceeded 90% when at least one source directly faced the sensor. Conversely, more stable conditions (D-class and below) showed reduced performance, though POD remained above 50%. Notably, the wide 95% confidence interval—particularly under stable conditions—reflects greater variability and uncertainty in detection performance under such scenarios. This underscores the challenge of consistent detection when dispersion is limited or plume-sensor alignment is suboptimal.

"In panel b, POD increases with relative distance from the source up to around 125 meters, after which it plateaus near 100%. This trend may result from enhanced plume overlap and alignment at mid-range distances, which improves detectability. Shorter distances could reduce POD due to narrow or offset plumes missing the sensor. The confidence interval narrows as distance increases, suggesting more consistent detection performance when sensors are positioned further downwind.

"In panel c Increasing the relative height of the sensor above the source also improves POD, with the highest detection probabilities

| | | |
|---|---|---|
| | occurring at the maximum tested height of 6 meters. This likely results from improved vertical intersection with the plume, especially under unstable conditions where the plume ascends more aggressively. The narrowing CI at higher sensor heights suggests increased detection reliability when sensors are elevated." | |
| Figure 7 Panel a: The stability classes are sorted so that the POD increases from low to high. This needs to be stated directly; otherwise this plot is misleading. | True. We propose to add this text: "In Panel (a), stability classes are intentionally arranged by increasing median POD rather than meteorological order to illustrate the relationship between atmospheric stability and detection performance clearly." | |
| Figure 7 Panel b: the 100 mark on the horizontal axis is not in the correct order. I think this is also plotted so that the POD is strictly increasing, which is misleading. The horizontal axis should be increasing distances. | We propose to replace that graph with this one, which is closer to the reviewer's description:  | |
| Figure 8: Is one pixel selected as the source location estimate? Or is the entire red swath the source estimate? The red region covers the sources, but it also covers much of the rest of the site. How do you pick a source location from the concentration maps shown in this figure? | It depends on the specific hot spot map we are looking at, but it can be between 1-4 potential source locations. We are simply showing one example of the Grandient Indicator procedural steps. If it continues, the persistence of points over the site area will retain potential candidate points. We usually don't search beyond a footprint length exceeding base expectations. For example, using a sensor at 4 meters in height, we | |

| | | |
|---|---|---|
| | would not expect to see anything outside a 40-200 meter distance, so anything outside that will be treated as an artifact.
This could be added to the text: "In the Gradient Indicator (GI) maps, red regions indicate areas with high gradient strength across the modeled footprint domain. These do not represent a definitive source location but rather a set of candidate source regions. The final source estimate is typically selected from among the most persistent high-gradient locations, often corresponding to one to four points. We limit the search space based on our expectations of footprint length to limit false detections, so for example, a 4 m height sensor indicates maximum source attribution distance of ~ 40–200 m downwind. Any detection further than this is considered a possible artifact and was dropped out of the selection." | |
| The POD values in Tab 3 don't seem to line up with the values in Fig 7 panel C. This needs to be checked. | Thanks for pointing this out. There was an issue with the x-axis height difference. The new graph will be added to the script.
 | |
| The localization accuracy (LA) acronym is defined twice in the abstract | Thank you for catching this. We propose to fix this. | |
| Methane is defined as "CH4" twice in the first paragraph of the introduction, | We propose to fix this. | |

| | | |
|---|---|---|
| but the authors continue to use "methane" throughout the paper. | | |
| L53: Jia et al. 2023 has been published: https://doi.org/10.1038/s41598-025-99491-x | We propose to fix this. | |
| L59: There is no Daniels et al. 2022 in the list of references. Do the authors mean to say Daniels et al. 2023 or Daniels et al. 2024? | We apologize for the typo. It is a 2024 paper. Corrections will be made to the manuscript. | |
| L71: The Gaussian puff and plume models studied in Jia et al. 2025 are not back-trajectory methods. They are forward models that simulate the transport of methane from the source to the sensor. | We propose to update our text to read: "Jia et al. (2025) explored the application of Gaussian plume and puff models as progressive diffusion techniques for $CH_4$ transport and found them to perform well in assessing complex release events. In comparison, simple back-trajectory approaches as used in commercial CEMS are often incapable of providing the horizontal/spatial scale or time resolution necessary to accurately attribute source events in these types of scenarios (Daniels et al. 2023)." | |
| L91-98: This discussion would be clearer if the authors first provided a clear definition of both "concentration footprint" and "flux footprint." | This added to the script "In order to have a more precise definition, a flux footprint refers to the surface area from which contributions to the measured flux at a given point, typically above the surface, are derived. This area integrates the effects of sources and sinks that influence the net flux of a scalar quantity due to turbulent transport (Kljun et al., 2015). Conversely, a concentration footprint describes the upwind area contributing to the observed concentration at a measurement point, like satellite imagery from above. Conceptually, it represents where the sampled air has come | |

| | from and how much each location contributes to the measured concentration. (Levin et al., 2020)." | |
|---|---|---|
| L113: "localization" is defined here but used multiple times earlier in the introduction. It should be defined at its first use. | Thank you for pointing that out. The definition can be added at its first use. | |
| L170: Need to provide some information about your coordinate system before you say things like the "x and y directions." | This has been responded to before in another comment. | |
| L205: GDM not defined | We propose to add: Gaussian Dispersion Model (GDM) | |
| Eq 1 is missing a plus sign | We propose to fix this | |
| Figure 2 needs a legend for the x's and the red dashed line. | Figures will be modified accordingly | |
| L297: A more precise definition of POD should be used. It looks like there is a Repeated sentence here as well. | We propose to add the following text: "The most important term used here is Probability of Detection (POD), a metric that describes how likely a monitoring system is to detect the presence of an emission when one occurs successfully." | |
| Your definition of localization accuracy (LA) is more commonly referred to as "positive predictive value" or "precision. " It might be better to use these more common phrases. | This can be fixed as described. | |
| In the FNF equation, it is not clear what the $n\_c$ refers to. It would be better to write out exactly how $n\_c$ is calculated (e.g., $n\_tp + n\_fn$), as is done with the other equations. | We would propose to add this. | |